# SUBJECT-SPECIFIC DEEP NEURAL NETWORKS FOR COUNT DATA WITH HIGH-CARDINALITY CATEGORICAL FEATURES

## ABSTRACT

There is a growing interest in subject-specific predictions using deep neural networks (DNNs) because real-world data often exhibit correlations, which has been typically overlooked in traditional DNN frameworks. In this paper, we propose a novel hierarchical likelihood learning framework for introducing gamma random effects into the Poisson DNN, so as to improve the prediction performance by capturing both nonlinear effects of input variables and subject-specific cluster effects. The proposed method simultaneously yields maximum likelihood estimators for fixed parameters and best unbiased predictors for random effects by optimizing a single objective function. This approach enables a fast end-to-end algorithm for handling clustered count data, which often involve high-cardinality categorical features. Furthermore, state-of-the-art network architectures can be easily implemented into the proposed h-likelihood framework. As an example, we introduce multi-head attention layer and a sparsemax function, which allows feature selection in high-dimensional settings. To enhance practical performance and learning efficiency, we present an adjustment procedure for prediction of random parameters and a method-of-moments estimator for pretraining of variance component. Various experiential studies and real data analyses confirm the advantages of our proposed methods.

## 1 INTRODUCTION

Deep neural networks (DNNs), which have been proposed to capture the nonlinear relationship between input and output variables (LeCun et al., 2015; Goodfellow et al., 2016), provide outstanding marginal predictions for independent outputs. However, in practical applications, it is common to encounter correlated data with high-cardinality categorical features, which can pose challenges for DNNs. While the traditional DNN framework overlooks such correlation, random effect models have emerged in statistics to make subject-specific predictions for correlated data. Lee & Nelder (1996) proposed hierarchical generalized linear models (HGLMs), which allow the incorporation of random effects from an arbitrary conjugate distribution of generalized linear model (GLM) family.

Both DNNs and random effect models have been successful in improving prediction accuracy of linear models but in different ways. Recently, there has been a rising interest in combining these two extensions. Simchoni & Rosset (2021; 2023) proposed the linear mixed model neural network for continuous (Gaussian) outputs with Gaussian random effects, which allow explicit expressions for likelihoods. Lee & Lee (2023) introduced the hierarchical likelihood (h-likelihood) approach, as an extension of classical likelihood for Gaussian outputs, which provides an efficient likelihood-based procedure. For non-Gaussian (discrete) outputs, Tran et al. (2020) proposed a Bayesian approach for DNNs with normal random effects using the variational approximation method (Bishop & Nasrabadi, 2006; Blei et al., 2017). Mandel et al. (2023) used a quasi-likelihood approach (Breslow & Clayton, 1993) for DNNs, but the quasi-likelihood method has been criticized for its poor prediction accuracy. Lee & Nelder (2001) proposed the use of Laplace approximation to have approximate maximum likelihood estimators (MLEs). Although Mandel et al. (2023) also applied Laplace approximation for DNNs, their method ignored many terms in the second derivatives due to computational expense, which could lead to inconsistent estimations (Lee et al., 2017). Therefore, a new approach is desired for non-Gaussian DNNs to obtain the MLEs for fixed parameters.

Clustered count data are widely encountered in various fields (Roulin & Bersier, 2007; Henderson & Shimakura, 2003; Thall & Vail, 1990; Henry et al., 1998), which often involve high-cardinality categorical features, i.e., categorical variables with a large number of unique levels or categories, such as subject ID or cluster name. However, to the best of our knowledge, there appears to be no available source code for the subject-specific Poisson DNN models. In this paper, we introduce Poisson-gamma DNN for the clustered count data and derive the h-likelihood that simultaneously provides MLEs of fixed parameters and best unbiased predictors (BUPs) of random effects. In contrast to the ordinary HGLM and DNN framework, we found that the local minima can cause poor prediction when the DNN model contains subject-specific random effects. To resolve this issue, we propose an adjustment to the random effect prediction that prevents from violation of the constraints for identifiability. Additionally, we introduce the method-of-moments estimators for pretraining the variance component. It is worth emphasizing that incorporating state-of-the-art network architectures into the proposed h-likelihood framework is straightforward. As an example, we implement a feature selection method with multi-head attention.

In Section 2 and 3, we present the Poisson-gamma DNN and derive its h-likelihood, respectively. In Section 4, we present the algorithm for online learning, which includes an adjustment of random effect predictors, pretraining variance components, and feature selection method using multi-head attention layer. In Section 5, we provide experimental studies to compare the proposed method with various existing methods. The results of the experimental studies clearly show that the proposed method improves predictions from existing methods. In Section 6, real data analyses demonstrate that the proposed method has the best prediction accuracy in various clustered count data. Proofs for theoretical results are derived in Appendix. Source codes are included in Supplementary Materials.

## 2 MODEL DESCRIPTIONS

### 2.1 POISSON DNN

Let $y_{ij}$ denote a count output and $\mathbf{x}_{ij}$ denote a $p$-dimensional vector of input features, where the subscript $(i, j)$ indicates the $j$th outcome of the $i$th subject (or cluster) for $i = 1, ..., n$ and $j = 1, ..., q_i$. For count outputs, Poisson DNN (Rodrigo & Tsokos, 2020) gives the marginal predictor,

$$\eta_{ij}^m = \log \mu_{ij}^m = \text{NN}(\mathbf{x}_{ij}; \mathbf{w}, \boldsymbol{\beta}) = \sum_{k=1}^{p_L} g_k(\mathbf{x}_{ij}; \mathbf{w})\beta_k + \beta_0, \tag{1}$$

where $\mu_{ij}^m = \text{E}(Y_{ij}|\mathbf{x}_{ij})$ is the marginal mean, $\text{NN}(\mathbf{x}_{ij}; \mathbf{w}, \boldsymbol{\beta})$ is the neural network predictor, $\boldsymbol{\beta} = (\beta_0, \beta_1, ..., \beta_{p_L})^T$ is the vector of weights and bias between the last hidden layer and the output layer, $g_k(\mathbf{x}_{ij}; \mathbf{w})$ denotes the $k$-th node of the last hidden layer, and $\mathbf{w}$ is the vector of all the weights and biases before the last hidden layer. Here the bold face stands for vectors and matrices, while non-bold face stands for scalars. In Poisson DNN, inverse of the log function, $\exp(\cdot)$, becomes the activation function of the output layer. Poisson DNNs allow highly nonlinear relationship between input and output variables, but only provide the marginal predictions for $\mu_{ij}^m$. Thus, Poisson DNN can be viewed as an extension of Poisson GLM with $\eta_{ij}^m = \mathbf{x}_{ij}^T \boldsymbol{\beta}$.

### 2.2 POISSON-GAMMA DNN

To allow subject-specific prediction into the model (1), we propose the Poisson-gamma DNN,

$$\eta_{ij}^c = \log \mu_{ij}^c = \text{NN}(\mathbf{x}_{ij}; \mathbf{w}, \boldsymbol{\beta}) + \mathbf{z}_{ij}^T \mathbf{v}, \tag{2}$$

where $\mu_{ij}^c = \text{E}(Y_{ij}|\mathbf{x}_{ij}, v_i)$ is the conditional mean, $\text{NN}(\mathbf{x}_{ij}; \mathbf{w}, \boldsymbol{\beta})$ is the marginal predictor of the Poisson DNN (1), $\mathbf{v} = (v_1, ..., v_n)^T$ is the vector of random effects from the log-gamma distribution, and $\mathbf{z}_{ij}$ is a vector from the model matrix for random effects, representing the high-cardinality categorical features. The conditional mean $\mu_{ij}^c$ can be formulated as

$$\mu_{ij}^c = \exp\{\text{NN}(\mathbf{x}_{ij}; \mathbf{w}, \boldsymbol{\beta})\} \cdot u_i,$$

where $u_i = \exp(v_i)$ is the gamma random effect.

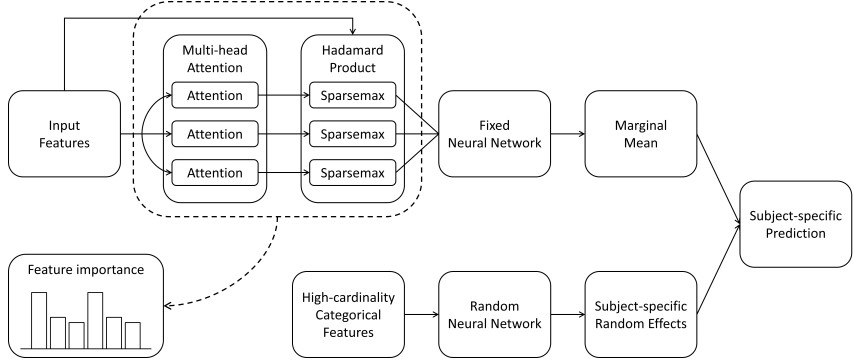

Figure 1: An example of the proposed model architecture. The input features are denoted by $\mathbf{x}_{ij}$ and the high-cardinality categorical features are denoted by $\mathbf{z}_{ij}$ in the proposed model (2).

Note here that, for any $\epsilon \in \mathbb{R}$, the model (2) can be expressed as

$$\log \mu_{ij}^c = \sum_{k=1}^{p_L} g_k(\mathbf{x}_{ij}; \mathbf{w})\beta_k + \beta_0 + v_i = \sum_{k=1}^{p_L} g_k(\mathbf{x}_{ij}; \mathbf{w})\beta_k + (\beta_0 - \epsilon) + (v_i + \epsilon),$$

or equivalently, for any $\delta = \exp(\epsilon) > 0$,

$$\mu_{ij}^c = \exp\left\{\text{NN}(\mathbf{x}_{ij}; \mathbf{w}, \boldsymbol{\beta})\right\} \cdot u_i = \exp\left\{\text{NN}(\mathbf{x}_{ij}; \mathbf{w}, \boldsymbol{\beta}) - \log \delta\right\} \cdot (\delta u_i),$$

which leads to an identifiability problem. Thus, it is necessary to place constraints on either the fixed parts $\text{NN}(\mathbf{x}_{ij}; \mathbf{w}, \boldsymbol{\beta})$ or the random parts $u_i$. Lee & Lee (2023) developed subject-specific DNN models with Gaussian outputs, imposing the constraint $\text{E}(v_i) = 0$, which is common for normal random effects. For Poisson-gamma DNNs, we use the constraints $\text{E}(u_i) = \text{E}(\exp(v_i)) = 1$ for subject-specific prediction of count outputs. The use of constraints $\text{E}(u_i) = 1$ has an advantage that the marginal predictions for multiplicative model can be directly obtained, because

$$\mu_{ij}^m = \text{E}[\text{E}(Y_{ij}|\mathbf{x}_{ij}, u_i)] = \text{E}\left[\exp\left\{\text{NN}(\mathbf{x}_{ij}; \mathbf{w}, \boldsymbol{\beta})\right\} \cdot u_i\right] = \exp\left\{\text{NN}(\mathbf{x}_{ij}; \mathbf{w}, \boldsymbol{\beta})\right\}.$$

Thus, we employ $v_i = \log u_i$ to the model (2), where $u_i \sim \text{Gamma}(\lambda^{-1}, \lambda^{-1})$ with $\text{E}(u_i) = 1$ and $\text{var}(u_i) = \lambda$. By allowing two separate output nodes, the Poisson-gamma DNN provides both marginal and subject-specific predictions,

$$\widehat{\mu}_{ij}^m = \exp(\widehat{\eta}_{ij}^m) = \exp\left\{\text{NN}(\mathbf{x}_{ij}; \widehat{\mathbf{w}}, \widehat{\boldsymbol{\beta}})\right\} \quad \text{and} \quad \widehat{\mu}_{ij}^c = \exp\left\{\text{NN}(\mathbf{x}_{ij}; \widehat{\mathbf{w}}, \widehat{\boldsymbol{\beta}}) + \mathbf{z}_{ij}^T \widehat{\mathbf{v}}\right\},$$

where the hats denote the predicted values. Subject-specific prediction can be achieved by multiplying the marginal mean predictor $\widehat{\mu}_{ij}^m$ and the subject-specific predictor of random effect $\widehat{u}_i = \exp(\widehat{v}_i)$. Note here that

$$\text{var}(Y_{ij}|\mathbf{x}_{ij}) = \text{E}\left\{\text{var}(Y_{ij}|\mathbf{x}_{ij}, v_i)\right\} + \text{var}\left\{\text{E}(Y_{ij}|\mathbf{x}_{ij}, v_i))\right\} \geq \text{E}\left\{\text{var}(Y_{ij}|\mathbf{x}_{ij}, v_i)\right\},$$

where $\text{var}\{\text{E}(Y_{ij}|\mathbf{x}_{ij}, v_i)\}$ represents the between-subject variance and $\text{E}\{\text{var}(Y_{ij}|\mathbf{x}_{ij}, v_i)\}$ represents the within-subject variance. To enhance the predictions, Poisson DNN improves the marginal predictor $\mu_{ij}^m = \text{E}(Y_{ij}|\mathbf{x}_{ij}) = \text{E}\{\text{E}(Y_{ij}|\mathbf{x}_{ij}, v_i)\}$ by allowing highly nonlinear function of $\mathbf{x}$, whereas Poisson-gamma DNN further uses the conditional predictor $\mu_{ij}^c = \text{E}(Y_{ij}|\mathbf{x}_{ij}, v_i)$, eliminating between-subject variance. Figure 1 illustrates an example of the proposed model architecture including feature selection in Section 4.

## 3 CONSTRUCTION OF H-LIKELIHOOD

For subject-specific prediction via random effects, it is important to define the objective function for obtaining exact MLEs of fixed parameters $\boldsymbol{\theta} = (\mathbf{w}, \boldsymbol{\beta}, \lambda)$. In the context of linear mixed models, Henderson et al. (1959) proposed to maximize the joint density with respect to fixed and random parameters. However, it cannot yield MLEs of variance components. There have been various

attempts to extend joint maximization schemes with different justifications (Gilmour et al., 1985; Harville & Mee, 1984; Schall, 1991; Breslow & Clayton, 1993; Wolfinger, 1993), but failed to obtain simultaneously the exact MLEs of all fixed parameters and BUPs of random parameters by optimizing a single objective function. It is worth emphasizing that defining joint density requires careful consideration because of the Jacobian term associated with the random parameters. For $\boldsymbol{\theta}$ and $\mathbf{u}$, an extended likelihood (Lee et al., 2017) can be defined as

$$\ell_e(\boldsymbol{\theta}, \mathbf{u}) = \log f_{\mathbf{y},\mathbf{u}}(\mathbf{y}, \mathbf{u}; \boldsymbol{\theta}) = \sum_{i,j} \log f_{y|u}(y_{ij}|u_i; \mathbf{w}, \boldsymbol{\beta}) + \sum_i \log f_u(u_i; \lambda). \tag{3}$$

However, a nonlinear transformation $v_i = v(u_i)$ of random effects $u_i$ leads to different extended likelihood due to the Jacobian terms:

$$\ell_e(\boldsymbol{\theta}, \mathbf{v}) = \log f_{\mathbf{y},\mathbf{v}}(\mathbf{y}, \mathbf{v}; \boldsymbol{\theta}) = \sum_{i,j} \log f_{y|v}(y_{ij}|v_i; \mathbf{w}, \boldsymbol{\beta}) + \sum_i \log f_v(v_i; \lambda)$$

$$= \sum_{i,j} \log f_{y|u}(y_{ij}|u_i; \mathbf{w}, \boldsymbol{\beta}) + \sum_i \log f_u(u_i; \lambda) + \sum_i \log \left| \frac{du_i}{dv_i} \right| \neq \ell_e(\boldsymbol{\theta}, \mathbf{u}).$$

The two extended likelihoods $\ell_e(\boldsymbol{\theta}, \mathbf{u})$ and $\ell_e(\boldsymbol{\theta}, \mathbf{v})$ lead to different estimates, raising the question on how to obtain the true MLEs. In Poisson-gamma HGLMs, Lee & Nelder (1996) proposed the use of $\ell_e(\boldsymbol{\theta}, \mathbf{v})$ that can give MLEs for $\boldsymbol{\beta}$ and BUPs for $\mathbf{u}$ by the joint maximization. However, their method could not give MLE for the variance component $\lambda$. In this paper, we derive the new h-likelihood whose joint maximization simultaneously yields MLEs of the whole fixed parameters including the variance component $\lambda$, BUPs of the random effects $\mathbf{u}$, and conditional expectations $\boldsymbol{\mu}^c = (\mu_{11}^c, ..., \mu_{n,q_n}^c)^T$.

Suppose that $\mathbf{v}^* = (v_1^*, ..., v_n^*)^T$ is a transformation of $\mathbf{v}$ such that

$$v_i^* = v_i \cdot \exp\{-c_i(\boldsymbol{\theta}; \mathbf{y}_i)\},$$

where $c_i(\boldsymbol{\theta}; \mathbf{y}_i)$ is a function of some parameters in $\boldsymbol{\theta}$ and $\mathbf{y}_i = (y_{i1}, ..., y_{iq_i})^T$ for $i = 1, 2, ..., n$. Let $c(\boldsymbol{\theta}; \mathbf{y}) = \sum_{i=1}^n c_i(\boldsymbol{\theta}; \mathbf{y}_i)$. Then we define the h-likelihood as

$$h(\boldsymbol{\theta}, \mathbf{v}) \equiv \log f_{\mathbf{v}^*|\mathbf{y}}(\mathbf{v}^*|\mathbf{y}; \mathbf{w}, \boldsymbol{\beta}, \lambda) + \log f_{\mathbf{y}}(\mathbf{y}; \mathbf{w}, \boldsymbol{\beta}, \lambda) = \ell_e(\boldsymbol{\theta}, \mathbf{v}) + c(\boldsymbol{\theta}; \mathbf{y}), \tag{4}$$

if the joint maximization of $h(\boldsymbol{\theta}, \mathbf{v})$ leads to MLEs of all the fixed parameters and BUPs of the random parameters. A sufficient condition for $h(\boldsymbol{\theta}, \mathbf{v})$ to yield exact MLEs of all the fixed parameters in $\boldsymbol{\theta}$ is that $f_{\mathbf{v}^*|\mathbf{y}}(\widetilde{\mathbf{v}}^*|\mathbf{y})$ is independent of $\boldsymbol{\theta}$, where $\widetilde{\mathbf{v}}^*$ is the mode,

$$\widetilde{\mathbf{v}}^* = \operatorname*{argmax}_{\mathbf{v}^*} h(\boldsymbol{\theta}, \mathbf{v}) = \operatorname*{argmax}_{\mathbf{v}^*} \log f_{\mathbf{v}^*|\mathbf{y}}(\mathbf{v}^*|\mathbf{y}; \mathbf{w}, \boldsymbol{\beta}, \lambda).$$

For the proposed model, Poisson-gamma DNN, we found that the following function

$$c_i(\boldsymbol{\theta}; \mathbf{y}_i) = (y_{i+} + \lambda^{-1}) + \log \Gamma(y_{i+} + \lambda^{-1}) - (y_{i+} + \lambda^{-1}) \log(y_{i+} + \lambda^{-1})$$

satisfies the sufficient condition,

$$\log f_{\mathbf{v}^*|\mathbf{y}}(\widetilde{\mathbf{v}}^*|\mathbf{y}) = \sum_{i=1}^n \log f_{v^*|\mathbf{y}}(\widetilde{v}_i^*|\mathbf{y}) = \sum_{i=1}^n \left\{ \log f_{v|\mathbf{y}}(\widetilde{v}_i|\mathbf{y}) + c_i(\boldsymbol{\theta}; \mathbf{y}_i) \right\} = 0,$$

where $y_{i+} = \sum_{j=1}^{q_i} y_{ij}$ is the sum of outputs in $\mathbf{y}_i$ and $\widetilde{v}_i = \widetilde{v}_i^* \cdot \exp\{c_i(\boldsymbol{\theta}; \mathbf{y}_i)\}$. Since $c_i(\boldsymbol{\theta}; \mathbf{y}_i)$ only depends on $\lambda$ and $y_{i+}$, we denote $c(\boldsymbol{\theta}; \mathbf{y}) = \sum_i c_i(\boldsymbol{\theta}; \mathbf{y}_i)$ by $c(\lambda; \mathbf{y}) = \sum_i c_i(\lambda; y_{i+})$. Then, the h-likelihood at mode $h(\boldsymbol{\theta}, \widetilde{\mathbf{v}})$ becomes the classical (marginal) log-likelihood,

$$\ell(\boldsymbol{\theta}; \mathbf{y}) = \log f_{\mathbf{y}}(\mathbf{y}; \mathbf{w}, \boldsymbol{\beta}, \lambda) = \log \int_{\mathbb{R}^n} \exp\{\ell_e(\boldsymbol{\theta}, \mathbf{v})\} d\mathbf{v}. \tag{5}$$

Thus, joint maximization of the h-likelihood (4) provides exact MLEs for the fixed parameters $\boldsymbol{\theta}$, including the variance component $\lambda$. BUPs of $\mathbf{u}$ and $\boldsymbol{\mu}^c$ can be also obtained from the h-likelihood,

$$\widetilde{\mathbf{u}} = \exp(\widetilde{\mathbf{v}}) = \mathrm{E}(\mathbf{u}|\mathbf{y}) \quad \text{and} \quad \widetilde{\boldsymbol{\mu}}^c = \exp(\widetilde{\mathbf{v}}) \cdot \exp\{\mathrm{NN}(\mathbf{x}; \mathbf{w}, \boldsymbol{\beta})\} = \mathrm{E}(\boldsymbol{\mu}^c|\mathbf{y}).$$

The proof and technical details for the theoretical results are derived in Appendix A.1.

---

**Algorithm 1** Learning algorithm for Poisson-gamma DNN via h-likelihood

---

**Input:** $\mathbf{x}_{ij}, \mathbf{z}_{ij}$
**for** epoch $= 0$ **to** pretrain epochs **do**
   Train $\mathbf{w}$, $\boldsymbol{\beta}$ and $\mathbf{v}$ by minimizing the negative h-likelihood.
   Compute method-of-moments estimator of $\lambda$.
   Adjust the random effect predictors.
**end for**
**for** epoch $= 0$ **to** train epochs **do**
   Train all the fixed and random parameters by minimizing the negative h-likelihood.
   Adjust the random effect predictors.
**end for**
Compute MLE of $\lambda$.

---

## 4    Learning algorithm for Poisson-gamma DNN models

In this section, we introduce the h-likelihood learning framework for handling the count data with high-cardinality categorical features. We decompose the negative h-likelihood loss for online learning and propose additional procedures to improve the prediction performance. The entire learning algorithm of the proposed method is briefly described in Algorithm 1.

### 4.1    Loss function for online learning

The proposed Poisson-gamma DNN can be trained by optimizing the negative h-likelihood loss,

$$\text{Loss} = -h(\boldsymbol{\theta}, \mathbf{v}) = -\log f_{\mathbf{y}|\mathbf{v}}(\mathbf{y}|\mathbf{v}; \mathbf{w}, \boldsymbol{\beta}) - \log f_{\mathbf{v}}(\mathbf{v}; \lambda) - c(\lambda; \mathbf{y}),$$

which is a function of the two separate output nodes $\mu_{ij}^m = \text{NN}(\mathbf{x}_{ij}; \mathbf{w}, \boldsymbol{\beta})$ and $v_i = \mathbf{z}_{ij}^T \mathbf{v}$. To apply online stochastic optimization methods, the proposed loss function is expressed as

$$\text{Loss} = \sum_{i,j} \left[ -y_{ij} \left( \log \mu_{ij}^m + v_i \right) + e^{v_i} \mu_{ij}^m - \frac{v_i - e^{v_i} - \log \lambda}{q_i \lambda} + \frac{\log \Gamma(\lambda^{-1})}{q_i} - c_i(\lambda; y_{i+}) \right]. \quad (6)$$

### 4.2    Random Effect Adjustment

While DNNs often encounter local minima, Dauphin et al. (2014) claimed that in ordinary DNNs, local minima may not necessarily result in poor predictions. In contrast to HGLM and DNN, we observed that the local minima can lead to poor prediction when the network reflects subject-specific random effects. In Poisson-gamma DNNs, we impose the constraint $\text{E}(u_i) = 1$ for identifiability, because for any $\delta > 0$,

$$\mu_{ij}^c = \exp\left\{\text{NN}(\mathbf{x}_{ij}; \mathbf{w}, \boldsymbol{\beta})\right\} \cdot u_i = \left[\exp\left\{\text{NN}(\mathbf{x}_{ij}; \mathbf{w}, \boldsymbol{\beta}) - \log \delta\right\}\right] \cdot (\delta u_i).$$

However, in practice, Poisson-gamma DNNs often end with local minima that violate the constraint. To prevent poor prediction due to local minima, we introduce an adjustment to the predictors of $u_i$,

$$\widehat{u}_i \leftarrow \frac{\widehat{u}_i}{\frac{1}{n} \sum_{i=1}^{n} \widehat{u}_i} \quad \text{and} \quad \widehat{\beta}_0 \leftarrow \widehat{\beta}_0 + \log\left(\frac{1}{n} \sum_{i=1}^{n} \widehat{u}_i\right) \quad (7)$$

to satisfy $\sum_{i=1}^{n} \widehat{u}_i/n = 1$. The following theorem shows that the proposed adjustment improves the local h-likelihood prediction. The proof is given in Appendix A.2.

**Theorem 1** *In Poisson-gamma DNNs, suppose that $\widehat{\beta}_0$ and $\widehat{u}_i$ are estimates of $\beta_0$ and $u_i$ such that $\sum_{i=1}^{n} \widehat{u}_i/n = 1 + \epsilon$ for some $\epsilon \in \mathbb{R}$. Let $\widehat{u}_i^*$ and $\widehat{\beta}_0^*$ be the adjusted estimators in (7). Then,*

$$h(\widehat{\boldsymbol{\theta}}^*, \widehat{\mathbf{v}}^*) \geq h(\widehat{\boldsymbol{\theta}}, \widehat{\mathbf{v}}),$$

*and the equality holds if and only if $\epsilon = 0$, where $\widehat{\boldsymbol{\theta}}$ and $\widehat{\boldsymbol{\theta}}^*$ are vectors of the same fixed parameter estimates but with different $\widehat{\beta}_0$ and $\widehat{\beta}_0^*$ for $\beta_0$, respectively.*

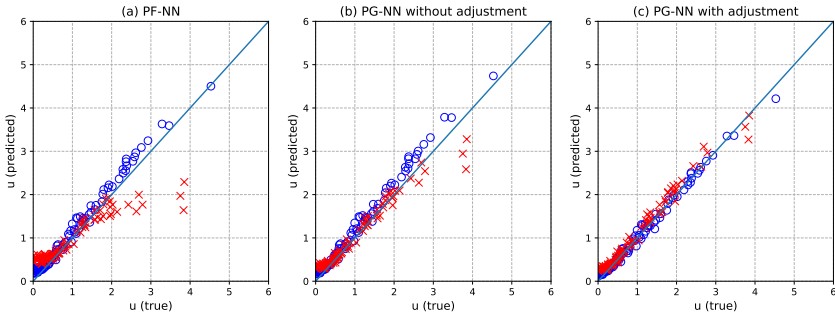

Figure 2: Predicted values of $u_i$ from two replications (marked as o and x for each) when $u_i$ is generated from the Gamma distribution with $\lambda = 1$, $n = 100$, $q = 100$.

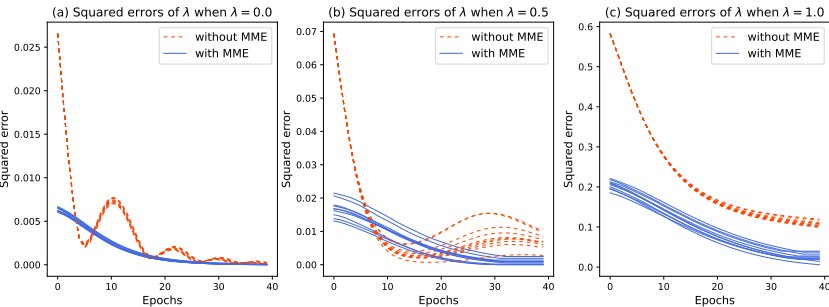

Figure 3: Learning curve for the variance component $\lambda$ when (a) $\lambda = 0$, (b) $\lambda = 0.5$, and (c) $\lambda = 1$.

Theorem 1 shows that the adjustment (7) improves the random effect prediction. According to our experience, even though limited, this adjustment becomes important, especially when the cluster size is large. Figure 2 is the plot of $\widehat{u}_i$ against the true $u_i$ under $(n, q) = (100, 100)$ and $\lambda = 1$. Figure 2 (a) shows that the use of fixed effects for subject-specific effects (PF-NN) produces poor prediction of $u_i$. Figure 2 (b) and (c) show that the use of random effects for subject-specific effects (PG-NN) improves the subject-specific prediction, and the proposed adjustment improves it further.

### 4.3 PRETRAINING VARIANCE COMPONENTS

We found that the MLE for variance component $\lambda = \text{var}(u_i)$ could be sensitive to the choice of initial value, giving a slow convergence. We propose the use of method-of-moments estimator (MME) for pretraining $\lambda$,

$$\widehat{\lambda} = \left[\frac{1}{n}\sum_{i=1}^{n}(\widehat{u}_i - 1)^2\right]\left[\frac{1}{2} + \sqrt{\frac{1}{4} + \frac{n\sum_i^n \widehat{\mu}_{i+}^{-1}(\widehat{u}_i - 1)^2}{\{\sum_i^n(\widehat{u}_i - 1)^2\}^2}}\right], \tag{8}$$

where $\widehat{\mu}_{i+} = \sum_{j=1}^{q_i}\widehat{\mu}_{ij}^m$. Convergence of the MME (8) is shown in Appendix A.3. Figure 3 shows that the proposed pretraining accelerates the convergence in various settings. In Appendix A.4, we demonstrate an additional experiments for verifying the consistency of $\widehat{\lambda}$ of the proposed method.

### 4.4 FEATURE SELECTION IN HIGH DIMENSIONAL SETTINGS

Feature selection methods can be easily implemented to the proposed PG-NN. For examples, we implemented permutation importance and attention-based feature selection using the multi-head attention layer with sparsemax function (Martins & Astudillo, 2016; Škrlj et al., 2020; Arik & Pfister, 2021),

$$\text{sparsemax}(\mathbf{a}) = \underset{\mathbf{b}\in\Delta^{p-1}}{\arg\min} ||\mathbf{b} - \mathbf{a}||^2,$$

where $\Delta^{p-1} = \{\mathbf{b} \in \mathbb{R}^p : \mathbf{1}^T\mathbf{b} = 1, \mathbf{b} \geq 0\}$ and $p$ is the number of features. As a high dimensional setting, we generate input features $x_{kij}$ from $N(0, 1)$ for $k = 1, ..., 100$, including 10 genuine features ($k \leq 10$) and 90 irrelevant features ($k > 10$). The output $y_{ij}$ is generated from $\text{Poi}(\mu_{ij}^c)$ with the mean model,

$$\mu_{ij}^c = u_i \cdot \exp\left[0.2\left\{1 + \cos x_{1ij} + \cdots + \cos x_{6ij} + (x_{7ij}^2 + 1)^{-1} + \cdots + (x_{10ij}^2 + 1)^{-1}\right\}\right],$$

where $u_i = e^{v_i}$ is generated from $\text{Gamma}(2, 2)$. The number of subjects (the cardinality of the categorical feature) is $n = 10^4$. The number of repeated measures (cluster size) is set to be $q = 20$, which is smaller than number of features $p = 100$. We employed a multi-layer perceptron with 20-10-10 number of nodes and three-head attention layer for feature selection. Figures in Appendix A.5 show that both the attention-based feature selection and permutation importance can identify the genuine features. The main difference between them is that the attention-based feature selection can achieve sparsity during training, whereas the permutation importance is a post-hoc procedure.

## 5 EXPERIMENTAL STUDIES

To investigate the performance of the Poisson-gamma DNN, we conducted experimental studies. The five input variables $\mathbf{x}_{ij} = (x_{1ij}, ..., x_{5ij})^T$ are generated from the AR(1) process with autocorrelation $\rho = 0.5$ for each $i = 1, ..., n$ and $j = 1, ..., q$. The random effects are generated from either $u_i \sim \text{Gamma}(\lambda^{-1}, \lambda^{-1})$ or $v_i \sim N(0, \lambda)$ where $v_i = \log u_i$. Since the random effect is unobserved, the distributional assumption for random effects is often hard to check in practice. Thus, we considered the case where $v_i \sim N(0, \lambda)$ is the true distribution to show that our PG-modeling is robust against misspecification of random effect distribution. When $\lambda = 0$, the conditional mean $\mu_{ij}^c$ is identical to the marginal mean $\mu_{ij}^m$. The output variable $y_{ij}$ is generated from $\text{Poisson}(\mu_{ij}^c)$ with

$$\mu_{ij}^c = u_i \cdot \exp\left[0.2\left\{1 + \cos x_{1ij} + \cos x_{2ij} + \cos x_{3ij} + (x_{4ij}^2 + 1)^{-1} + (x_{5ij}^2 + 1)^{-1}\right\}\right].$$

Results are based on the 100 sets of simulated data. The data consist of $q = 10$ observations for $n = 1000$ subjects. For each subject, 6 observations are assigned to the training set, 2 are assigned to the validation set, and the remaining 2 are assigned to the test set.

For comparison, we consider the following models.

- **P-GLM**    Classic Poisson GLM for count outputs using `R`.
- **N-NN**    Conventional DNN for continuous outputs.
- **P-NN**    Poisson DNN for count outputs.
- **PN-GLM**    Poisson-normal HGLM using `lme4` (Bates et al., 2015) package in `R`.
- **PG-GLM**    Poisson-gamma HGLM using the proposed method.
- **NF-NN**    Conventional DNN with fixed subject-specific effects for continuous outputs.
- **NN-NN**    DNN with normal random effects for continuous outputs (Lee & Lee, 2023).
- **PF-NN**    Conventional Poisson DNN with fixed subject-specific effects for count outputs.
- **PG-NN**    The proposed Poisson-gamma DNN for count outputs.

To evaluate the prediction performances, we consider the root mean squared Pearson error (RMSPE)

$$\text{RMSPE} = \sqrt{\frac{1}{N}\sum_{i,j}\frac{(y_{ij} - \widehat{\mu}_{ij})^2}{V(\widehat{\mu}_{ij})}},$$

where $\text{Var}(y_{ij}|u_i) = \phi V(\mu_{ij})$ and $\phi$ is a dispersion parameter of GLM family. For Gaussian outputs, the RMSPE is identical to the ordinary root mean squared error, since $\phi = \sigma^2$ and $V(\mu_{ij}) = 1$. For Poisson outputs, $\phi = 1$ and $V(\mu_{ij}) = \mu_{ij}$. P-GLM, N-NN, and P-NN give marginal predictions $\widehat{\mu}_{ij} = \widehat{\mu}_{ij}^m$, while the others give subject-specific predictions $\widehat{\mu}_{ij} = \widehat{\mu}_{ij}^c$. N-NN, NF-NN, and NN-NN are models for continuous outputs, while the others are for count outputs. For NF-NN and PF-NN, predictions are made by maximizing the conditional likelihood $\log f_{\mathbf{y}|\mathbf{v}}(\mathbf{y}|\mathbf{v}; \mathbf{w}, \boldsymbol{\beta})$. For PN-GLM, PG-GLM, NN-NN, and PG-NN, subject-specific predictions are made by maximizing the h-likelihood.

Table 1: Mean and standard deviation of test RMSPEs of simulation studies over 100 replications. G(0) implies the absence of random effects, i.e., $v_i = 0$ for all $i$. Bold numbers indicate the minimum.

| Model | Distribution of random effects ($\lambda$) | | | | |
|---|---|---|---|---|---|
| | G(0) & N(0) | G(0.5) | G(1) | N(0.5) | N(1) |
| P-GLM | 1.046 (0.029) | 1.501 (0.055) | 1.845 (0.085) | 1.745 (0.113) | 2.818 (0.467) |
| N-NN | 1.013 (0.018) | 1.473 (0.042) | 1.816 (0.074) | 1.713 (0.097) | 1.143 (0.432) |
| P-NN | **1.011 (0.018)** | 1.470 (0.042) | 1.812 (0.066) | 1.711 (0.099) | 1.161 (0.440) |
| PN-GLM | 1.048 (0.029) | 1.112 (0.033) | 1.115 (0.035) | 1.124 (0.030) | 1.152 (0.034) |
| PG-GLM | 1.048 (0.020) | 1.123 (0.027) | 1.106 (0.023) | 1.139 (0.026) | 1.161 (0.028) |
| NF-NN | 1.152 (0.029) | 1.301 (0.584) | 1.136 (0.311) | 1.241 (1.272) | 1.402 (0.298) |
| NN-NN | 1.020 (0.020) | 1.121 (0.026) | 1.209 (0.067) | 1.256 (0.097) | 2.773 (0.384) |
| PF-NN | 1.147 (0.025) | 1.135 (0.029) | 1.128 (0.027) | 1.129 (0.024) | 1.128 (0.027) |
| PG-NN | 1.016 (0.019) | **1.079 (0.024)** | **1.084 (0.023)** | **1.061 (0.022)** | **1.085 (0.026)** |

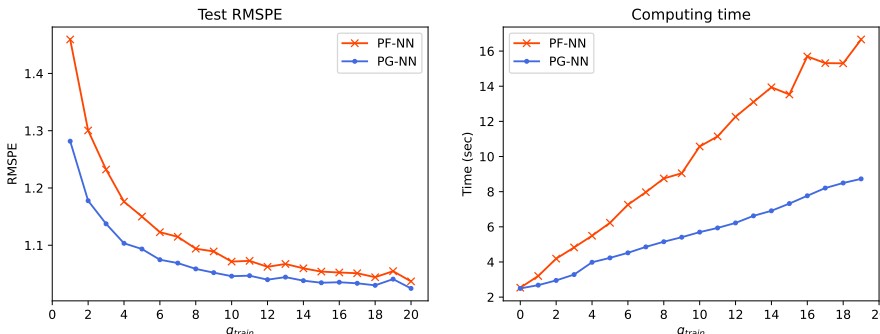

Figure 4: Average of test RMSPEs and computing times from 100 repetitions.

PN-GLM is the generalized linear mixed model with random effects $v_i \sim N(0, \lambda)$. Current statistical software for PN-GLM and PG-GLM (`lme4` and `dhglm`) provide approximate MLEs using Laplace approximation. The proposed learning algorithm can yield exact MLEs for PG-GLM, using solely the input and output layers while excluding the hidden layers. Among various methods for NN-NN (Tran et al., 2020; Mandel et al., 2023; Simchoni & Rosset, 2021; 2023; Lee & Lee, 2023), we applied the state-of-the-art method proposed by Lee & Lee (2023). For all DNNs, we employed a standard multi-layer perceptron (MLP) consisting of 3 hidden layers with 10 neurons and leaky ReLU activation function. All the DNNs and PG-GLMs were implemented in Python using Keras (Chollet et al., 2015) and TensorFlow (Abadi et al., 2015). We applied the Adam optimizer with a learning rate of 0.001 and an early stopping process based on the validation loss while training the DNNs. NVIDIA Quadro RTX 6000 were used for computations.

Table 1 shows the mean and standard deviation of test RMSPEs from the experimental studies. When the true model does not have random effects (G(0) and N(0)), PG-NN is comparable to P-NN without random effects, which should perform the best (marked by the bold face) in terms of RMSPE. N-NN (P-NN) without random effects is also better than NF-NN and NN-NN (PF-NN and PG-NN) with random effects. When the distribution of random effects is correctly specified (G(0.5) and G(1)), PG-NN performs the best in terms of RMSPE, with p-values<0.001 from the Wilcoxon signed-rank test (paired Wilcoxon test), used in Zhang et al. (2015; 2017); Xiong et al. (2019). Even when the distribution of random effects is misspecified (N(0.5), N(1)), PG-NN still performs the best with p-values<0.001. This result is in accordance with the simulation results of McCulloch & Neuhaus (2011), namely, in GLMMs, the prediction accuracy is little affected for violations of the distributional assumption for random effects: see similar performances of PN-GLM and PG-GLM.

It has been known that handling the high-cardinality categorical features as random effects has advantages over handling them as fixed effects (Lee et al., 2017), especially when the cardinality of

Table 2: Test RMSPEs of real data analyses. Bold numbers indicate the minimum values.

| Model | Longitudinal data | | | Clustered data | |
|-------|-------|-----|-------|------|--------|
| | Epilepsy | CD4 | Bolus | Owls | Fruits |
| P-GLM | 1.520 | 6.115 | 2.110 | 2.493 | 6.203 |
| N-NN | 2.119 | 8.516 | 1.982 | 6.695 | 5.953 |
| P-NN | 1.712 | 6.830 | 2.354 | 6.689 | 5.954 |
| PN-GLM | 1.242 | 3.422 | 1.727 | 2.492 | 7.128 |
| PG-GLM | 1.229 | 4.424 | 1.714 | 2.602 | 6.159 |
| NF-NN | 1.750 | 6.921 | 1.718 | 2.473 | 6.566 |
| NN-NN | 1.770 | 7.640 | 1.727 | 2.463 | 6.234 |
| PF-NN | 1.238 | 3.558 | 1.816 | 2.496 | 5.906 |
| PG-NN | **1.135** | **3.513** | **1.677** | **2.427** | **5.901** |

categorical feature $n$ is close to the total sample size $N = \sum_{i=1}^{n} q_i$, i.e., the number of observations in each category (cluster size $q$) is relatively small. Thus, to emphasize the advantages of PG-NN over PF-NN in high-cardinality categorical features, we consider an experiment with cluster size $q_{\text{train}}$ varying from 1 to 20 with $\lambda = 0.2$ and $n = 1000$. Figure 4 show that PG-NN has uniformly smaller RMSPE and less computation time than PF-NN. The difference of RMSPEs becomes severe when $q_{\text{train}}$ is small, i.e., the ratio of cardinality and total sample size $n/N$ becomes close to 1. Therefore, the proposed method enhances subject-specific predictions as the cardinality of categorical features becomes high.

## 6 REAL DATA ANALYSIS

To investigate the prediction performance of clustered count outputs in practice, we examined the following five real datasets: Epilepsy data (Thall & Vail, 1990), CD4 data (Henry et al., 1998), Bolus data (Henderson & Shimakura, 2003), Owls data (Roulin & Bersier, 2007), and Fruits data (Banta et al., 2010). For all DNNs, a standard MLP with one hidden layer of 10 neurons and a sigmoid activation function were employed. For longitudinal data (Epilepsy, CD4, Bolus), the last observation for each patient was used as the test set without cross-validation, as in Tran et al. (2020). For clustered data (Owls, Fruits), we present the average of RMSPEs from 10-fold cross-validation. Table 2 shows that the proposed PG-NN has the smallest RMSPEs. P-values are presented in Appendix A.6. Throughout the datasets, P-GLM performs better than P-NN, implying that non-linear model does not improve the linear model in the absence of subject-specific random effects. Meanwhile, in the presence of subject-specific random effects, PG-NN is always preferred to PG-GLM. The results imply that introducing subject-specific random effects in DNNs can help to identify the nonlinear effects of the input variables. Therefore, while DNNs are widely recognized for improving predictions in independent datasets, introducing subject-specific random effects could be necessary for DNNs to improve their predictions in correlated datasets with high-cardinality categorical features.

## 7 CONCLUDING REMARKS

When the data contains high-cardinality categorical features, introducing random effects into DNNs is advantageous. We develop subject-specific Poisson-gamma DNN for clustered count data. The h-likelihood enables a fast end-to-end learning algorithm using the single objective function. By introducing subject-specific random effects, DNNs can effectively identify the nonlinear effects of the input variables. Various state-of-the-art network architectures can be easily implemented into the h-likelihood framework, as we demonstrate with the feature selection based on multi-head attention.

With increasing interest in combining DNNs with statistical models, there has been theoretical developments, such as the identifiability of fixed effects for better interpretation (Rügamer, 2023). Since random effects have different nature compared to fixed effects, exploring the use of random effects in DNNs for better interpretability could open up interesting topics for future research.

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

# A  APPENDIX

## A.1  DERIVATION OF THE H-LIKELIHOOD

Maximizing $\log f_{\mathbf{v}|\mathbf{y}}(\mathbf{v}|\mathbf{y}; \mathbf{w}, \boldsymbol{\beta}, \lambda)$ with respect to $\mathbf{v} = (v_1, ..., v_n)^T$ yields,

$$\tilde{v}_i \equiv \underset{v_i}{\operatorname{argmax}} \log f_{v|\mathbf{y}}(v_i|\mathbf{y}_i; \mathbf{w}, \boldsymbol{\beta}, \lambda)$$

$$= \underset{v_i}{\operatorname{argmax}} \left[ \sum_{j=1}^{q_i} \left( y_{ij}v_i - \mu_{ij}^m e^{v_i} \right) + \frac{v_i - e^{v_i}}{\lambda} \right] = \log \left( \frac{y_{i+} + \lambda^{-1}}{\mu_{i+} + \lambda^{-1}} \right),$$

where $y_{i+} = \sum_{j=1}^{q_i} y_{ij}$ and $\mu_{i+} = \sum_{j=1}^{q_i} \mu_{ij}^m$. As derived in Section 3, define $c_i(\lambda; y_{i+})$ as

$$c_i(\lambda; y_{i+}) = (y_{i+} + \lambda^{-1}) + \log \Gamma(y_{i+} + \lambda^{-1}) - (y_{i+} + \lambda^{-1}) \log(y_{i+} + \lambda^{-1})$$

and consider a transformation,

$$v_i^* = v_i \cdot \exp\{-c_i(\lambda; y_{i+})\}.$$

Since the multiplier $\exp\{-c_i(\lambda; y_{i+})\}$ does not depend on $v_i$, we have

$$\tilde{v}_i^* \equiv \underset{v_i^*}{\operatorname{argmax}} \log f_{v^*|\mathbf{y}}(v_i^*|\mathbf{y}_i; \mathbf{w}, \boldsymbol{\beta}, \lambda) = \tilde{v}_i \cdot \exp\{-c_i(\lambda; y_{i+})\},$$

which leads to

$$\log f_{\mathbf{v}^*|\mathbf{y}}(\widetilde{\mathbf{v}}^*|\mathbf{y}) = \sum_{i=1}^{n} \left\{ \log f_{v|\mathbf{y}}(\widetilde{v}_i|\mathbf{y}_i) + c_i(\lambda; y_{i+}) \right\} = 0.$$

This satisfies the sufficient condition for the h-likelihood to give exact MLEs for fixed parameters,

$$\underset{\boldsymbol{\theta}}{\operatorname{argmax}} \, h(\boldsymbol{\theta}, \widetilde{\mathbf{v}}) = \underset{\boldsymbol{\theta}}{\operatorname{argmax}} \, \ell(\boldsymbol{\theta}; \mathbf{y}).$$

Furthermore, from the distribution of $u_i|\mathbf{y}_i$,

$$\widetilde{u}_i = \exp(\widetilde{v}_i) = \frac{y_{i+} + \lambda^{-1}}{\mu_{i+} + \lambda^{-1}} = \mathrm{E}(u_i|\mathbf{y}_i)$$

and

$$\widetilde{\mu}_{ij}^c = \exp(\widetilde{v}_i) \cdot \exp\{\mathrm{NN}(\mathbf{x}; \mathbf{w}, \boldsymbol{\beta})\} = \mu_{ij}^m \cdot \mathrm{E}(u_i|\mathbf{y}_i) = \mathrm{E}(\mu_{ij}^c|\mathbf{y}).$$

Therefore, maximizing the h-likelihood yields the BUPs of $u_i$ and $\mu_{ij}^c$.

## A.2  PROOF OF THEOREM 1.

The adjustment (7) transports

$$\widehat{u}_i^* = \widehat{u}_i/(1 + \epsilon) \quad \text{and} \quad \widehat{v}_i^* = \widehat{v}_i - \log(1 + \epsilon).$$

Since $(\widehat{\boldsymbol{\theta}}, \widehat{\mathbf{v}})$ and $(\widehat{\boldsymbol{\theta}}^*, \widehat{\mathbf{v}}^*)$ have the same conditional expectation $\widehat{\mu}_{ij}$, equation (4) yields

$$h(\widehat{\boldsymbol{\theta}}^*, \widehat{\mathbf{v}}^*) - h(\widehat{\boldsymbol{\theta}}, \widehat{\mathbf{v}}) = \sum_{i=1}^{n} \left[ \frac{\widehat{v}_i^* - \exp(\widehat{v}_i^*)}{\widehat{\lambda}} \right] - \sum_{i=1}^{n} \left[ \frac{\widehat{v}_i - \exp(\widehat{v}_i)}{\widehat{\lambda}} \right]$$

$$= \widehat{\lambda}^{-1} \left\{ \sum_{i=1}^{n} (\widehat{v}_i^* - \widehat{v}_i) - \sum_{i=1}^{n} (\widehat{u}_i^* - \widehat{u}_i) \right\}$$

$$= n\widehat{\lambda}^{-1} \{\epsilon - \log(1 + \epsilon)\} \geq 0,$$

and the equality holds if and only if $\epsilon = 0$. Thus, $h(\widehat{\boldsymbol{\theta}}^*, \widehat{\mathbf{v}}^*) \geq h(\widehat{\boldsymbol{\theta}}, \widehat{\mathbf{v}})$.

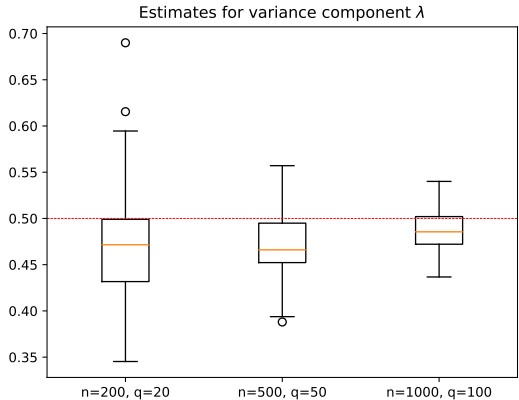

Figure 5: Box-plots of $\widehat{\lambda}$ when true $\lambda = 0.5$.

### A.3 Convergence of the method-of-moments estimator

As derived in Section A, for given $\lambda$ and $\mu_{i+}$, maximization of the h-likelihood leads to

$$\widehat{u}_i = \widehat{u}_i(\mathbf{y}_i) = \exp\left(\widehat{v}_i(\mathbf{y}_i)\right) = \frac{y_{i+} + \lambda^{-1}}{\mu_{i+} + \lambda^{-1}}.$$

Thus, $\mathrm{E}(\widehat{u}_i) = 1$ and $\mathrm{Var}(\widehat{u}_i) = \lambda\left\{1 - (\lambda\mu_{i+} + 1)^{-1}\right\}$. Define $d_i$ as

$$d_i = \frac{\widehat{u}_i - 1}{\sqrt{1 - (\lambda\mu_{i+} + 1)^{-1}}} = \frac{y_{i+} - \mu_{i+}}{\mu_{i+} + \lambda^{-1}} \sqrt{1 + \lambda^{-1}\mu_{i+}^{-1}}$$

to have $\mathrm{E}(d_i) = 0$ and $\mathrm{Var}(d_i) = \lambda$ for any $i = 1, ..., n$. Then, by the law of large numbers,

$$\frac{1}{n}\sum_{i=1}^{n} d_i^2 \to \mathrm{E}(d_i^2) = \mathrm{Var}(d_i) + \mathrm{E}(d_i)^2 = \lambda.$$

Note here that

$$\frac{1}{n}\sum_{i=1}^{n} d_i^2 = \left\{\frac{1}{n}\sum_{i=1}^{n}(\widehat{u}_i - 1)^2\right\} + \frac{1}{\lambda}\left\{\frac{1}{n}\sum_{i=1}^{n}\frac{(\widehat{u}_i - 1)^2}{\mu_{i+}}\right\}.$$

Then, solving the following equation,

$$\lambda - \left\{\frac{1}{n}\sum_{i=1}^{n}(\widehat{u}_i - 1)^2\right\} - \frac{1}{\lambda}\left\{\frac{1}{n}\sum_{i=1}^{n}\frac{(\widehat{u}_i - 1)^2}{\widehat{\mu}_{i+}}\right\} = 0,$$

leads to an estimate $\widehat{\lambda}$ in (8) and $\widehat{\lambda} \to \lambda$ as $n \to \infty$.

### A.4 Consistency of variance component estimator

By jointly maximizing the proposed h-likelihood, we obtain MLEs for fixed parameters and BUPs for random parameters. Thus, we can directly apply the consistency of MLEs under usual regularity conditions. To verify this consistency in PG-NN, we present the boxplots in Figure 5 for the variance component estimator $\widehat{\lambda}$ from 100 repetitions with true $\lambda = 0.5$. The number of clusters $n$ and cluster size $q$ are set to be (200, 20), (500, 50), and (1000,100). Figure 5 provide conclusive evidence confirming the consistency of the variance component estimator within the PG-NN framework.

### A.5 Feature selection methods

we implemented the permutation importance method and attention-based feature selection method to our experiment in Section 4.4. Figure 6 shows the average attention score of each feature from 100 repetitions to summarize all the results. Figure 7 shows the attention score from one repetition. Figure 8 shows the permutation score from one repetition.

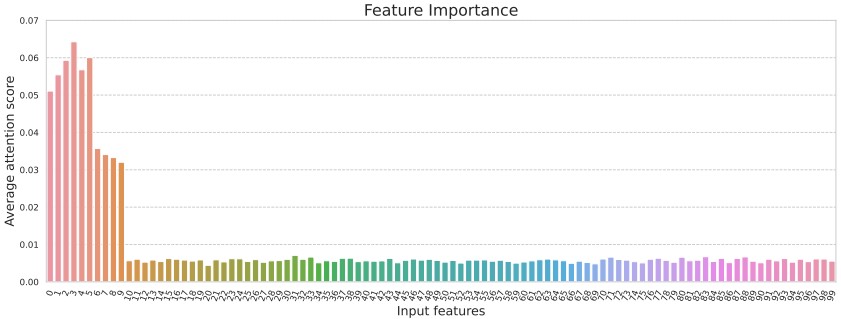

Figure 6: Average attention scores from 100 repetitions.

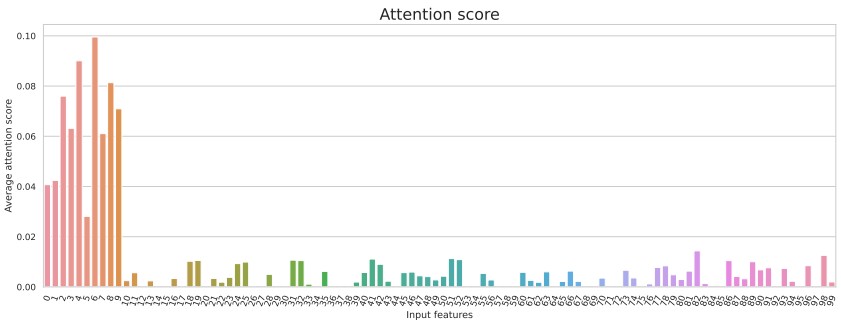

Figure 7: Attention scores from one repetition.

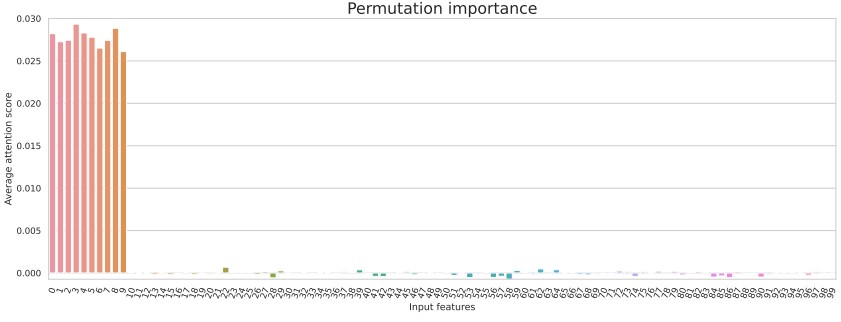

Figure 8: Permutation importance from one repetition.

A.6 REAL DATA ANALYSES

In this section, we provide details of the datasets in Section 6.

- **Epilepsy data:** Epilepsy data are reported by Thall & Vail (1990) from a clinical trial of $n = 59$ patients with epilepsy. The data contain $N = 236$ observations with $q_i = 4$ repeated measures from each patient and $p = 4$ input variables.

- **CD4 data:** CD4 data are from a study of AIDS patients with advanced immune suppression, reported by Henry et al. (1998). The data contain $N = 4612$ observations from $n = 1036$ patients with $q_i \geq 2$ repeated measurements and $p = 4$ input variables.

- **Bolus data:** Bolus data are from a clinical trial following abdominal surgery for $n = 65$ patients with $q_i = 12$ repeated measurements, reported in Henderson & Shimakura (2003). The data have $N = 780$ observations with $p = 2$ input variables.

- **Owls data:** Owls data are reported by Roulin & Bersier (2007), which can be found in the R package `glmmTMB` (Brooks et al., 2023). The data contain $N = 599$ observations and $n = 27$ nests with $p = 3$ input variables. The cluster size $q_i$ in each nest varies from 4 to 52.

- **Fruits data:** Fruits data are reported in Banta et al. (2010). The data have $N = 625$ observations clustered by $n = 24$ types of maternal seed family with $p = 3$ input variables. The cluster size $q_i$ varies from 11 to 47.

For clustered data (Owls, Fruits), we performed 10-fold cross-validation and Wilcoxon signed-rank test for validating the advantage of PG-NN, as in Zhang et al. (2015; 2017); Xiong et al. (2019). (Hence there is no p-value of PG-NN.) Table 3 presents the mean RMSPEs and p-values for testing whether the RMSPE of PG-NN is less than that of other methods. In Owls data, though the p-values are marginal (0.05~0.1) in some cases, the proposed PG-NN significantly (p-values<0.05) outperforms the other methods. In Fruits data, PG-NN has the smallest RMSPEs but the difference is not significant except for N-NN and NN-NN. Note here that P-GLM was the best in Fruits data without cross-validation, but the cross-validation clarifies that PG-NN has better overall RMSPE.

|  | Owls data | | Fruits data | |
| --- | --- | --- | --- | --- |
|  | RMSPE | p-value | RMSPE | p-value |
| P-GLM | 2.493 | 0.032 | 6.203 | 0.053 |
| PN-GLM | 6.695 | 0.001 | 5.953 | 0.116 |
| PG-GLM | 6.689 | 0.001 | 5.954 | 0.138 |
| N-NN | 2.492 | 0.065 | 8.943 | 0.001 |
| NF-NN | 2.602 | 0.080 | 6.837 | 0.001 |
| NN-NN | 2.473 | 0.032 | 7.308 | 0.001 |
| P-NN | 2.463 | 0.042 | 6.277 | 0.053 |
| PF-NN | 2.496 | 0.007 | 5.906 | 0.216 |
| PG-NN | 2.427 |  | 5.893 |  |

Table 3: Average RMSPEs and p-values from 10-fold cross-validation

