# OpenReview forum: "Subject-specific Deep Neural Networks for Count Data with High-cardinality Categorical Features"
_ICLR.cc/2024/Conference — ICLR 2024 Conference Withdrawn Submission_

### Official Review · Reviewer_HXxH · 2023-10-28

**Soundness:** 2 fair
**Presentation:** 2 fair
**Contribution:** 2 fair
**Rating:** 3
**Confidence:** 3

**Summary:**

The paper proposes a way to learn a deep neural network (DNN) with Poisson outcome distribution and Gamma-distributed random effects in one joint optimization approach using the h-likelihood framework. Apart from the derivation of a constant to obtain both maximum likelihood estimates for the fixed effects model part and best unbiased predictions for the random effects, the authors also using a method-of-moments approach to circumvent non-identifiability.

**Strengths:**

- Significance: Including random effects in neural networks is an important research topic that has not been fully explored
- Originality: The paper seems to be the first to tackle Poisson DNNs with random effects in this way

**Weaknesses:**

- Significance:
    + It's not clear how much better the given idea works compared to other approaches
    + Figure 4 shows that there is no true sparsity achieved, and separability between relevant and irrelavant features might depend on the size of the gap; using an actual l1-penalty via [overparametrization](https://arxiv.org/abs/2307.03571) could help in this case.
    + Statements in 4.2
        * Theorem 1: Maybe I have missed something, but AFAIU it's trivial that the likelihood will improve once you shift fixed information from random to fixed effects as both the likelihood and "prior" will increase (this is also common knowledge in mixed models). I would thus see this as part of Section 2.2.
        * Online or post-hoc corrections like the one in (7) probably work, but why not directly encode the constraint by encoding the information in z (i.e., using the design matrix $\tilde{Z} = Z - 1_n 1_n^\top Z / n$ instead of Z where Z is the matrix stacking all observations $z_{ij}$)?
    + Section 3: It's unclear to me why it is important to align $\ell_e(\theta,v)$ and $\ell_e(\theta, u)$ as both are simply two different model assumptions. The reason this might not be clear is that the same $f_\theta$ is used, yet not explicitly defined. More specifically, if $u_i$ follows a Gamma distribution, then $f_\theta$ should be the density of a Gamma distribution. But using **the same** Gamma density to evaluate the likelihood of a transformed random variable $v(u_i)$ does not make sense. If $v$ is $log(\cdot)$, then $v_i$ yields values that can be negative while the Gamma distribution is only defined for positive values. If 2) the authors mean that one transforms $v_i$ and then plugs in $u_i$ into a Gamma density (or the other way around), then there should not be any additional Jacobian term as the log-Gamma distribution is exactly defined in a way such that $exp(v_i) \sim Gamma$.
- Clarity: The math is not always clear, e.g.,
    + $f_\theta$ in (3) is never explicitly defined if I am not mistaken
    + bold/non-bold symbols are not used in a consistent manner (e.g. $\boldsymbol{\mu}^m$ and $\boldsymbol{\mu}^c$ in Section 2)
    + I assume $f_\theta(y|u)$ does not actually depend on $\theta$ directly? (in all fairness, this is also not very clear in many papers by Lee and authors which are cited for this)
    + in Section 4.4 the typical sparsemax notation is adopted but not aligned with the previous notation (it does not get clear what $\textbf{z}$ and $\textbf{p}$ are in the context of the paper -- in particular as z is already used by the "random slope features", but the lines below suggest that the section talks about selection of fixed effects) and also contains typos (e.g., it should say "genuine features $(K < 10)$" I assume, not $k < 10$).
- Limited empirical evidence: The simulation study is rather limited with
    + fixed n, number of features, etc.
    + no comparison against other hglm approaches (e.g. hglm package in R)
    + no comparison with REML approaches (?)

  and sometimes not very clear
    +  why simulate $v_i \sim N(0, \lambda)$ if it's clear that $log (u_i)$ won't follow a Gamma distribution?
    +  why does Table 1 say "Distribution of random effects" if the content is RMSPE values? Actually measuring the distribution quality would give more insights into the estimation performance.
- Quality: the phrasing is a bit akward at several places (e.g. "However, it could not yield MLE for the variance component $\lambda$")

**Questions:**

1. Can you comment on my comment on Theorem 1?
2. Can you clarify the idea behind Section 3 and the two different likelihoods?
3. The authors talk about obtaining "MLEs" and "BUPs". How is the unbiasedness for random effects defined (as there are multiple ways to do this) and can the authors prove that unbiasedness is something that can be achieved, even in the case of a "highly non-convex" neural network likelihood (that influences the random effects at least indirectly)?
4. Can z also contain information from x and if yes, how is identifiability ensured in this case (cf. [Ruegamer, 2023](https://proceedings.mlr.press/v202/rugamer23a.html))?
5. How do REML approaches fit into the picture?
6. Would an EM-based approach for NNs like in [Xiong et al., 2019](https://openaccess.thecvf.com/content_CVPR_2019/papers/Xiong_Mixed_Effects_Neural_Networks_MeNets_With_Applications_to_Gaze_Estimation_CVPR_2019_paper.pdf) be a meaningful alternative?

---

> ### Author Response · Authors · 2023-11-20
> **Response to Reviewer HXxH**
>
> We greatly appreciate you for your insightful and interesting comments, which provided diverse perspectives on this paper. We have prepared responses for your questions and revised the manuscript.
> We supplemented our experimental studies with increasing sample size in Section 5 and permutation importance in Section 4.4 and Appendix A.5. We provided p-values from Wilcoxon signed-rank test for comparing RMSPEs in experimental studies and real data analyses. We have supplemented the explanation on the notation and corrected some typos. In Section 7, we added remarks on future research. We are delighted to have enhanced our paper and hope that it will meet your expectations as well.
>
> ### **Question 1**
>
> (1) It could be a part of theoretical section, as you mentioned, but this could be a practical issue, because Theorem 1 justifies the adjustment process. If we can achieve the global optimum, as in linear models, then this kind of adjustment would not be necessary. However, since the global optimum may not be guaranteed in DNNs, there could be room for improving the fixed effects and random effects. Theorem 1 shows that our adjustment always reduce the loss function.
>
> (2) Due to the limited support for matrix representation in the OpenReview system, we are attaching a link for the screenshot of our response.
>
> * [Answer to Question 1-(2)](https://i.imgur.com/uwSwFxv.png)
>
> ### **Question 2 and clarity of notation**
>
> * Bold face stands for vectors and matrices and non-bold face stands for scalars.
>
> * $\boldsymbol{\theta}$ is the vector of all the fixed parameters, including all the weights $\mathbf{w}$ in neural network NN$(\cdot)$, fixed effects $\boldsymbol{\beta}$ from the last hidden layer to the output layer, and variance component $\lambda$ for the distribution of random effects. For the briefness of equation, we write the subscript $\boldsymbol{\theta}$ for dependency on any fixed parameters in $\boldsymbol{\theta}$, rather than describe a specific parameter for each function.
>
> * For notational convenience, $f_{\boldsymbol{\theta}}(\cdot)$ stands for a density function of the random variable in the parentheses, depending on some parameters in $\boldsymbol{\theta}$. For example, $f_{\boldsymbol{\theta}}(u) = f_{U}(u; \boldsymbol{\theta})$ denotes the probability density function of $u$, $f_{\boldsymbol{\theta}}(v) = f_{V}(v; \boldsymbol{\theta})$ denotes the probability density function of $v$, $f_{\boldsymbol{\theta}}(y,v) = f_{Y,V}(y,v; \boldsymbol{\theta})$ denotes the joint probability density function of $(y, v)$. As far as we know, this is conventional notation in statistics.
>
> * In Section 4.4, we changed $\mathbf{z}$ and $\mathbf{p}$ to $\mathbf{a}$ and $\mathbf{b}$, respectively. $K$ is corrected to $p$, which denotes the number of features. $k$ denotes the index of each feature. Thus, there are $p=100$ features including 10 genuine features $(k=1,...,10)$ and 90 irrelevant features $(k=11,...,100)$. We corrected our explanation in Section 4.4.
>
> ### **Question 3**
>
> We intended to say that theoretically the joint maximizer of h-likelihood is equivalent to the MLEs and BUPs of fixed and random parameters, respectively. However, as you mentioned, there is no guarantee to achieve the global maxima in practice. That is the reason why we proposed the use of an adjustment process in Theorem 1.
>
>
> ### **Question 4**
>
> Many thanks for introducing an interesting paper by Rügamer [1], which consider the identifiability problem in semi-structured networks (SSNs). Rügamer [1] considered SSNs with a late fusion of the structured and unstructured model,
> $$
> E(\mathbf{y}|\mathbf{X}, \mathbf{X}^*)
> = \boldsymbol{\eta}
> = \mathbf{X} \boldsymbol{\beta} + \mathbf{U} \boldsymbol{\gamma},
> $$
> where $\mathbf{X}$ is tabular features and $\mathbf{X}^*$ is additional features. Here, $\mathbf{U}$ are latent features learned in the penultimate layer of the neural network (i.e., $\mathbf{U}$ are latent features learned from $\mathbf{X}^*$), and $\boldsymbol{\gamma}$ are the weights from the connection between the last hidden and the output layer. The predictor $\boldsymbol{\eta}$ is combined with structured predictor $(\boldsymbol{\eta}^{\text{str}} = \mathbf{X}\boldsymbol{\beta})$ and non-structured predictor $(\boldsymbol{\eta}^{\text{unstr}} = \mathbf{X}\boldsymbol{\beta})$. In SSNs where both $\boldsymbol{\beta}$ and $\boldsymbol{\gamma}$ are fixed effects, Rügamer [1] studied identifiability of $\boldsymbol{\eta}^{\text{str}}$.
>
> In this paper, we are handling the identifiability between fixed and random effects (Section 2.2), which are quite different from that between two fixed effects $\boldsymbol{\beta}$ and $\boldsymbol{\gamma}$. We believe that, if $\boldsymbol{\gamma}$ were treated as random effects, it would satisfy the orthogonality properties for identifiability, which is very interesting topic, requesting a separate paper for extensive future research. We added a remark in Section 6 for your suggestion.

---

> ### Author Response · Authors · 2023-11-20
> **Response to Reviewer HXxH (continued)**
>
> ### **Question 5**
>
> REML approaches for linear mixed models (LMMs) can be implemented into DNNs by regarding the last hidden layer as the covariates of LMMs [2, 3].
>
> ### **Question 6**
>
> The variational EM approach [4] is another way for handling the random effects in DNNs. EM algorithm is composed of the E-step and M-step. It computes random effect predictors after estimating the fixed effects, including the variance components. On the other hand, the h-likelihood approach jointly optimizes the loss function, hence it only maximize the h-likelihood without an E-step. It can give more efficient and stable procedure. Lee and Lee [3] showed the advantage of joint optimization in their experimental studies.
>
> ### **Advantage of the given idea and limited empirical evidence**
>
> During the rebuttal period, we presented some additional experiments to clarify the advantage of the proposed method. First, we performed the Wilcoxon signed rank test for the RMSPEs, which is widely-used for comparing the prediction errors [4-6].
>
> * In experimental studies, PG-NN outperforms the other models with p-value$<$0.001 in all scenarios except for G(0), with the absence of random effects. In this case, P-NN performs the best.
> * Following Tran et al. [7, Section 6.2.4], we did not consider the cross-validation for longitudinal data (Epilepsy, CD4, Bolus), because the last observations (test data) were predicted by training the past data for each subject. For clustered data (Owls, Fruits), we present the average of RMSPEs from 10-fold cross-validation with the p-values from Wilcoxon signed rank test. The following table (Table 3 in Appendix A.6) shows the mean RMSPEs and p-values for testing whether the RMSPE of PG-NN is less than that of other methods. (Hence the p-value for PG-NN itself is not included in this table.) In Owls data, though the p-values are marginal (0.05$\sim$0.1) in some cases, the proposed PG-NN significantly (p-value$<$0.05) outperforms the other methods. In Fruits data, the PG-NN has the smallest RMSPEs but the difference is not significant except for N-NN and NN-NN. Note here that the P-GLM was the best in Fruits data without cross-validation, but the cross-validation clarifies that the PG-NN has better overall RMSPE.
>
> |Model|Owls/RMSPE|Owls/p-value|Fruits/RMSPE|Fruits/p-value|
> |---|---|---|---|---|
> |P-GLM |2.493|0.032|6.203|0.065|
> |PN-GLM|6.695|0.001|5.953|0.138|
> |PG-GLM|6.689|0.001|5.954|0.138|
> |N-NN  |2.492|0.065|7.128|0.001|
> |NF-NN |2.602|0.080|6.159|0.042|
> |NN-NN |2.473|0.032|6.566|0.003|
> |P-NN  |2.463|0.042|6.234|0.053|
> |PF-NN |2.496|0.007|5.906|0.161|
> |PG-NN |2.427|     |5.901|     |
>
> Furthermore, for handling extremely high-cardinality categorical features, it has been acknowledged that the random effect models have less degrees of freedom (less model complexity) than the fixed effect models [8, Chapter 6.5]. We presented an experiment with $q_{train}$ varying from 1 to 20 $(N=1,000 \sim 20,000)$. Figure 4 in Section 5 shows the average RMSPEs and computing times of standard PF-NN and the proposed PG-NN from 100 repetitions. The proposed PG-NN requires less computation time, while maintaining smaller RMSPE than standard PF-NN.
>
> * [Figure 4 (Link)](https://i.imgur.com/7YNF1xi.png)
>
>
> ### **Feature selection and L1-penalty**
>
> Due to page limitations, Figure 4 in the initial submission was moved to Figure 6 in Appendix A.5. This figure illustrates the average of the feature importance for 100 repetitions to summarize all the results. Figure 5 in Appendix A.5 shows the attention scores from one repetition, and we can recognize this attention-based feature selection can achieve the sparsity.
>
> Since the L1-penalty promotes sparsity by pushing the less important weights of a neural network towards zero, it induces sparsity in the solution of weights rather than in features. Therefore, the L1-penalty could be helpful for achieving the sparsity of individual weight, but in different context from the sparsity of an input feature itself, i.e., exclusion of all the weights from that feature.
>
> Additionally, we implemented the permutation importance for feature selection and added the result in Appendix A.5.
>
> ### **Comparison with \texttt{hglm} package in R**
>
> Actually, we tried hglm package in R for fitting PG-GLM, but we did not present it because it encountered the singularity problems. Furthermore, the hglm package cannot provide the MLE, because it uses Laplace approximation [3], whereas the new h-likelihood can provide the MLE without using any approximation.
>
> ### **Scenario with N$(\lambda)$**
> Since the random effect is unobserved, the distributional assumption for random effects is often hard to check in practice. Thus, it is important to show that our PG-modeling is robust against misspecification of random effect distribution. We considered the case where $v_i \sim N(0,\lambda)$ is the true distribution, which is one of the commonly used assumptions for random effects.

---

> > ### Author Response · Authors · 2023-11-20
> > **References in Response to Reviewer HXxH**
> >
> > ### **English**
> >
> > * "Distribution of random effects" in Table 4 is not the title of the table but represents the columns (N(0), N(0.5), etc.).
> > * We revised several typos and corrected the English. (e.g. "However, their method cannot give MLE for the variance component $\lambda$.")
> >
> > ### **References**
> >
> > [1] Rügamer, D. (2023). A New PHO-rmula for Improved Performance of Semi-Structured Networks. Proceedings of the 40th International Conference on Machine Learning, in Proceedings of Machine Learning Research 202:29291-29305 Available from https://proceedings.mlr.press/v202/rugamer23a.html.
> >
> > [2] Mandel, F., Ghosh, R. P., \& Barnett, I. (2023). Neural networks for clustered and longitudinal data using mixed effects models. Biometrics, 79(2), 711-721.
> >
> > [3] Lee, H., \& Lee, Y. (2023). H-Likelihood Approach to Deep Neural Networks with Temporal-Spatial Random Effects for High-Cardinality Categorical Features. Proceedings of the 40th International Conference on Machine Learning, in Proceedings of Machine Learning Research 202:18974-18987 Available from https://proceedings.mlr.press/v202/lee23k.html.
> >
> > [4] Xiong, Y., Kim, H. J., \& Singh, V. (2019). Mixed effects neural networks (menets) with applications to gaze estimation. In Proceedings of the IEEE/CVF conference on computer vision and pattern recognition (pp. 7743-7752).
> >
> > [5] Zhang, X., Sugano, Y., Fritz, M., \& Bulling, A. (2015). Appearance-based gaze estimation in the wild. In Proceedings of the IEEE conference on computer vision and pattern recognition (pp. 4511-4520).
> >
> > [6] Zhang, X., Sugano, Y., Fritz, M., \& Bulling, A. (2017). MIIGaze: Real-world dataset and deep appearance-based gaze estimation. IEEE transactions on pattern analysis and machine intelligence, 41(1), 162-175.
> >
> > [7] Tran, M. N., Nguyen, N., Nott, D., \& Kohn, R. (2020). Bayesian deep net GLM and GLMM. Journal of Computational and Graphical Statistics, 29(1), 97-113.
> >
> > [8] Lee, Y., Nelder, J. A., \& Pawitan, Y. (2017). Generalized linear models with random effects: unified analysis via H-likelihood (Vol. 153). CRC Press.

---

> > > ### Comment · Reviewer_HXxH · 2023-11-21
> > > **Response to Authors' Response**
> > >
> > > I thank the authors for their response, the additional materials provided as well as clarifying various aspects of their work. A couple of follow-up questions (if time still allows to answer):
> > >
> > > - Q1:
> > >     1. Does it really make a difference if it is a linear model or an NN? The two parts are additive (on the predictor level) and one (the random effect part) assumes a simple structure with mean of zero / of one. This identifiability problem is always present as long as you don't orthogonalize the two parts. Imho, it is the same problem as in generalized additive models, where the intercept is not identifiable without constraints. And then, per definition, anything that deviates from the mean assumption will decrease the "prior probability".
> > >     2. Correct. My proposed approach is different from the authors' approach, but it will ensure that the second term on the RHS of (2) in the paper is zero mean in sum and hence the NN part will capture the intercept If I am not mistaken. (Just as a side note)
> > > - Q2:
> > >     > Bold face stands for vectors and matrices and non-bold face stands for scalars.
> > >
> > >     Right. I was just mentioning this as a friendly pointer on areas where things can be improved -- because you actually did not use it consistently throughout the paper IIRC.
> > >
> > >     > [...]  As far as we know, this is conventional notation in statistics.
> > >
> > >     Again, just a friendly pointer. I would consider myself a statistician and I found it confusing / usually read papers that make it (more) explicit.
> > >
> > >
> > >     It also seems that the authors have missed or misunderstood my question to explain the idea of the different likelihoods in Section 3 ("why to align the two log-likelihoods").
> > >
> > > - Other comments:
> > >
> > >     > With the new h-likelihood, our procedure is better than hglm package in R."
> > >
> > >     That seems like a bold claim without any empirical evidence. Does this have theoretical justificiation?

---

> ### Author Response · Authors · 2023-11-22
>
> Thank you for your prompt feedback. Regarding your additional questions, we have prepared the following responses.
>
> ### **Question 1**
>
> #### **On the adjustment**
>
> (1) In linear models under the assumption $\text{E}(u_i)=1$, the random effect predictors automatically satisfy $\bar{u} = \sum_{i=1}^{n} \widehat{u}_i/n = 1$, since we achieve the global optimum. Thus, the adjustment and Theorem 1 would not be necessary for linear models. However, since we may not achieve the global optimum in DNNs, we introduced the adjustment to give $\bar{u}=1$.
>
> As you mentioned, the adjustment does not change $\widehat{\mu}_{ij}^c$, but it changes the loss function (negative h-likelihood). During training, it influences the estimate of the variance component $\lambda$, which in turn affects the predictions of random effects.
>
> (2) We believe that your idea, using
> $\tilde{\mathbf{Z}}=\mathbf{Z}-\mathbf{1}_n\mathbf{1}_n^T/n,$
> would be useful under the assumption $\text{E}(v_i)=0$,
> which is commonly assumed for normal random effects.
> However, for gamma random effects, we assume $\text{E}(u_i)=\text{E}(e^{v_i})=1$ in our PG modeling.
>
> * When $\text{E}(u_i)=1$, the fixed part can give the marginal mean,
> $$
> \text{E}(y_{ij}) = \mu_{ij}^m = \exp [ \text{NN}(\mathbf{x}_{ij}; \mathbf{w}, \boldsymbol{\beta}) ],
> $$
> as derived in Section 2.2.
>
> * On the other hand, when $\text{E}(v_i)=0$, the distribution of $u_i$ becomes
> $$
> u_i \sim \text{Gamma}\left(\alpha, e^{\psi(\alpha)}\right),
> $$
> with $\text{E}(u_i)=\alpha e^{-\psi(\alpha)}$ and $\text{var}(u_i)=\lambda=\alpha e^{-2\psi(\alpha)}$,
> where $\psi(\cdot)$ is the digamma function and $\alpha>0$, so that the marginal mean becomes
> $$
> \text{E}(y_{ij}) = \mu_{ij}^m = \exp [ \text{NN}(\mathbf{x}_{ij}; \mathbf{w}, \boldsymbol{\beta})] \cdot \alpha e^{-\psi(\alpha)},
> $$
> which depends on $\alpha$.
>
> #### **On the identifiability**
>
> For the identifiability problem, we regarded the last hidden layer of NN as given covariates, hence we simply adapted the results in HGLMs (Lee et al., 2017). It is worth noting that the identifiability of random effects has quite different nature from that of fixed effects. For example, consider a linear model,
> $$
> \mathbf{y} = \mathbf{X}\boldsymbol{\beta} + \mathbf{X}\boldsymbol{\gamma} + \mathbf{e},
> $$
> where $\mathbf{y}=(y_1, ..., y_N)^T$ is the response variable, $\mathbf{X}=(\mathbf{x}_1,...,\mathbf{x}_N)^T$ is the model matrix, $\mathbf{e} \sim N(\mathbf{0}, \sigma^2 \mathbf{I})$ is the random noise, $\boldsymbol{\beta}$ and $\boldsymbol{\gamma}$ are two different fixed effects. Then, the fixed effects $\boldsymbol{\beta}$ and $\boldsymbol{\gamma}$ are not identifiable, so the orthogonality condition is required for the identifiability of $\boldsymbol{\beta}$. On the other hand, if we consider the following model,
> $$
> \mathbf{y} = \mathbf{X}\boldsymbol{\beta} + \mathbf{X}\mathbf{v} + \mathbf{e},
> $$
> where $\mathbf{v}\sim N(\mathbf{0}, \lambda \mathbf{I})$ is normal random effects, then the fixed effect $\boldsymbol{\beta}$ is identifiable since $\widehat{\mathbf{v}} = \mathbf{0}$. Therefore, for the identifiability of random effects and related orthogonalization methods, a more thorough discussion in future research would be beneficial. We greatly appreciate your introduction of this interesting topic.

---

> > ### Author Response · Authors · 2023-11-22
> > **(continued)**
> >
> > ### **Question 2**
> >
> > Yes, we understood and we have modified the notations more explicitly. We hope that notational confusion is eliminated in the revised paper. The two extended likelihoods $\ell_e(\boldsymbol{\theta}, \mathbf{u})$ and $\ell_e(\boldsymbol{\theta}, \mathbf{v})$ are defined as
> > $$
> > \ell_e(\boldsymbol{\theta}, \mathbf{u})
> > = \log f_{\mathbf{y}, \mathbf{u}}(\mathbf{y}, \mathbf{u}; \mathbf{w}, \boldsymbol{\beta}, \lambda)
> > $$
> > $$
> > = \sum_{i,j} \left[ y_{ij} (\log \mu_{ij}^m + \log u_i) - \mu_{ij}^m u_i - \log y_{ij}! \right] + \sum_{i} \left[ \frac{\log u_i - u_i - \log \lambda}{\lambda} - \log \Gamma(\lambda^{-1}) - \log u_i \right]
> > $$
> > $$
> > = \sum_{i,j} \left[ y_{ij} (\log \mu_{ij}^m + v_i) - \mu_{ij}^m e^{v_i} - \log y_{ij}! \right] + \sum_{i} \left[ \frac{v_i - e^{v_i} - \log \lambda}{\lambda} - \log \Gamma(\lambda^{-1}) - v_i \right],
> > $$
> >
> > $$
> > \ell_e(\boldsymbol{\theta}, \mathbf{v}) = \log f_{\mathbf{y}, \mathbf{v}}(\mathbf{y}, \mathbf{v};\mathbf{w}, \boldsymbol{\beta}, \lambda)
> > $$
> > $$
> > = \sum_{i,j} \left[ y_{ij} (\log \mu_{ij}^m + v_i) - \mu_{ij}^m e^{v_i} - \log y_{ij}! \right] + \sum_{i} \left[ \frac{v_i - e^{v_i} - \log \lambda}{\lambda} - \log \Gamma(\lambda^{-1}) \right]
> > $$
> > $$
> > = \ell_e(\boldsymbol{\theta}, \mathbf{u}) + \sum_{i=1}^{n} v_i,
> > $$
> > where $\mu_{ij}^m=\exp[\text{NN}(\mathbf{x}_{ij};\mathbf{w},\boldsymbol{\beta})]$.
> > These two extended likelihoods are from the same model assumption (since $u_i \sim \text{Gamma}$ and $v_i=\log(u_i)\sim \text{log-Gamma}$ are equivalent), but lead to different optimization. Here we wanted to emphasize that the scale of random effect is important, because $\ell_e(\boldsymbol{\theta}, \mathbf{u})\neq \ell_e(\boldsymbol{\theta}, \mathbf{v})$ and they lead to different solutions under the same model, whereas the scale of fixed effect does not affect the solution.
> >
> > ### **Other comments**
> >
> > Many thanks for your comment. The sentence
> > > "Furthermore, the hglm package cannot provide MLE, because it uses Laplace approximation [3]. With the new h-likelihood, our procedure is better than hglm package in R."
> >
> > is replaced by
> >
> > >"Furthermore, the hglm package cannot provide the MLE, because it uses Laplace approximation [3], whereas the new h-likelihood can provide the MLE without using any approximation."
> >
> > In our experiments, the hglm package encountered singularity problems in the computation of Hessian, producing N/A values. Thus, we used the gradient-based learning algorithm in Tensorflow without computing the Hessian matrix.

---

> > > ### Comment · Reviewer_HXxH · 2023-11-22
> > >
> > > I thank the authors for the quick reply and clarifications! I understand now (I think) what the purpose of your way of adjustment $u$ and $\beta_0$ is. I will rethink all my points of criticism once again and take a look at the new results with a bit more time during the reviewer discussion phase.

---

### Official Review · Reviewer_ZRFo · 2023-11-01

**Soundness:** 4 excellent
**Presentation:** 3 good
**Contribution:** 3 good
**Rating:** 6
**Confidence:** 2

**Summary:**

This paper essentially extends subject-specific predictions to Poisson DNNs. It introduces Gamma random effects into Poisson DNNs and proposes a novel learning framework that can be applied when high-cardinality categorical features are present.  The experimental results across synthetic and real-world data suggest that in certain scenarios, employing this framework can yield advantageous results.

**Strengths:**

1) The theory checks out and the introduction of a method that simultaneously yields maximum likelihood estimators for fixed parameters and best unbiased predictors for random effects for Poisson DNNs is a valuable contribution.
2) The feature selection process for high cardinality categorical features is interesting and has valuable practical implications.

**Weaknesses:**

1) The synthetic experiments are rather limited. I would like to see how certain hyperparameters affect the experiments. For example, for any $x_{ij}$ only 5 features are chosen from an AR(1) process, is this choice standard? Why is the AR(1) process considered?
2) The improvements are marginal across both synthetic and real-world experiments.

**Questions:**

1) Could you please provide more examples of why Poisson DNN modelling might be advantageous compared to previous methods, especially in real-world experiments that you have considered?
2) At some point, after running a set of synthetic experiments, it is claimed that “Therefore, the proposed method enhances subject-specific predictions as the cardinality of categorical features becomes high.” However, this claim seems rather premature especially since only $q_{train} = 3$ and $q_{train} = 1$ have been considered. Could you please provide the RMSPE of PF-NN and PG-NN on a line chart where different values of $q_{train}$ are considered? This is needed to fully validate such a claim.

---

> ### Author Response · Authors · 2023-11-20
> **Response to Reviewer ZRFo**
>
> Thank you for your valuable feedback on our paper. We greatly appreciate your advice, which have contributed to enriching the content of our work. We have prepared responses for your questions and revised our manuscript. We supplemented our experimental studies with increasing sample size in Section 5 and permutation importance in Section 4.4 and Appendix A.5. We provided p-values from Wilcoxon signed-rank test for comparing RMSPEs in experimental studies and real data analyses. It has been our pleasure to improve our manuscript, and we hope it also meets with your approval.
>
> ### **Weakness 1**
>
> It is common in statistical literature to generate input features from AR(1) to exhibit the multicollinearity among them [1-3]. Though we could not investigate the impact for each hyperparameter, we tried to present various scenarios as possible, including the model misspecification, such as N(0.5) and N(1).
>
>
> ### **Weakness 2**
>
> Many thanks for a nice comment. For a statistical test, we performed the Wilcoxon signed-rank test for the RMSPEs, which is widely-used for comparing the prediction errors [4-6].
> * In experimental studies, PG-NN outperforms the other models with p-value$<$0.001 in all scenarios except for G(0), with the absence of random effects. In this case, P-NN performs the best.
> * Following Tran et al. [3, Section 6.2.4], we did not consider the cross-validation for longitudinal data (Epilepsy, CD4, Bolus), because the last observations (test data) were predicted by training the past data for each subject. For clustered data (Owls, Fruits), we present the average of RMSPEs from 10-fold cross-validation with the p-values from Wilcoxon signed rank test. The following table (Table 3 in Appendix A.6) shows the mean RMSPEs and p-values for testing whether the RMSPE of PG-NN is less than that of other methods. (Hence the p-value for PG-NN itself is not included in this table.) In Owls data, though the p-values are marginal (0.05$\sim$0.1) in some cases, the proposed PG-NN significantly (p-value$<$0.05) outperforms the other methods. In Fruits data, the PG-NN has the smallest RMSPEs but the difference is not significant except for N-NN and NN-NN. Note here that the P-GLM was the best in Fruits data without cross-validation, but the cross-validation clarifies that the PG-NN has better overall RMSPE.
>
> |Model|Owls/RMSPE|Owls/p-value|Fruits/RMSPE|Fruits/p-value|
> |---|---|---|---|---|
> |P-GLM |2.493|0.032|6.203|0.065|
> |PN-GLM|6.695|0.001|5.953|0.138|
> |PG-GLM|6.689|0.001|5.954|0.138|
> |N-NN  |2.492|0.065|7.128|0.001|
> |NF-NN |2.602|0.080|6.159|0.042|
> |NN-NN |2.473|0.032|6.566|0.003|
> |P-NN  |2.463|0.042|6.234|0.053|
> |PF-NN |2.496|0.007|5.906|0.161|
> |PG-NN |2.427|     |5.901|     |
>
> ### **Question 1**
>
> Poisson DNN is an extension of Poisson GLM (a standard regression model for count data) to highly non-linear model structures. The use of OLS for log-transformed count data is not satisfactory for analysis [7]. Though it was challenging to acquire large-sized datasets due to their confidential nature, count data are common in many domains since they reflect the number of occurrences of an outcome variable measured in unit period, area or volume. Here are some examples of application data: microbiome count data, CD4 count data, pandemic mortality data, insurance data, etc [8-12].
>
> ### **Question 2**
>
> Thank you for valuable comments. We presented an experiment with $q_{train}$ varying from 1 to 20 $(N=1,000 \sim 20,000)$. Figure 4 in Section 5 shows the average RMSPEs and computing times of standard PF-NN and the proposed PG-NN from 100 repetitions. The proposed PG-NN requires less computation time, while maintaining smaller RMSPE than standard PF-NN.
>
> * [Figure 4 (Link)](https://i.imgur.com/7YNF1xi.png)

---

> > ### Author Response · Authors · 2023-11-20
> > **References in Response to Reviewer ZRFo**
> >
> > ### **References**
> >
> > [1] Fan, J., \& Li, R. (2001). Variable selection via nonconcave penalized likelihood and its oracle properties. *Journal of the American statistical Association*, 96(456), 1348-1360.
> >
> > [2] Fan, J., Guo, S., \& Hao, N. (2012). Variance estimation using refitted cross-validation in ultrahigh dimensional regression. *Journal of the Royal Statistical Society Series B: Statistical Methodology*, 74(1), 37-65.
> >
> > [3] Tran, M. N., Nguyen, N., Nott, D., \& Kohn, R. (2020). Bayesian deep net GLM and GLMM. *Journal of Computational and Graphical Statistics*, 29(1), 97-113.
> >
> > [4] Zhang, X., Sugano, Y., Fritz, M., \& Bulling, A. (2015). Appearance-based gaze estimation in the wild. In *Proceedings of the IEEE conference on computer vision and pattern recognition*, 4511-4520.
> >
> > [5] Zhang, X., Sugano, Y., Fritz, M., \& Bulling, A. (2017). MIIGaze: Real-world dataset and deep appearance-based gaze estimation. *IEEE transactions on pattern analysis and machine intelligence*, 41(1), 162-175.
> >
> > [6] Xiong, Y., Kim, H. J., \& Singh, V. (2019). Mixed effects neural networks (menets) with applications to gaze estimation. In *Proceedings of the IEEE/CVF conference on computer vision and pattern recognition*, 7743-7752.
> >
> > [7] McCullagh, P. and Nelder, J. A. (1989). *Generalized Linear Models*. London: Chapman \& Hall.
> >
> > [8] Saison, J., Maucort Boulch, D., Chidiac, C., Demaret, J., Malcus, C., Cotte, L., ... \& Ferry, T. (2015, April). Increased Regulatory T-Cell Percentage Contributes to Poor CD4+ Lymphocytes Recovery: A 2-Year Prospective Study After Introduction of Antiretroviral Therapy. In *Open Forum Infectious Diseases*, 2(2), ofv063.
> >
> > [9] Zhang, X., Mallick, H., Tang, Z., Zhang, L., Cui, X., Benson, A. K., \& Yi, N. (2017). Negative binomial mixed models for analyzing microbiome count data. *BMC bioinformatics*, 18, 1-10.
> >
> > [10] Dong, M., Li, L., Chen, M., Kusalik, A., \& Xu, W. (2020). Predictive analysis methods for human microbiome data with application to Parkinson’s disease. *PLoS One*, 15(8), e0237779.
> >
> > [11] Yirga, A. A., Melesse, S. F., Mwambi, H. G., \& Ayele, D. G. (2020). Negative binomial mixed models for analyzing longitudinal CD4 count data. *Scientific reports*, 10(1), 16742.
> >
> > [12] Iyit, N., \& Sevim, F. (2023). A novel statistical modeling of air pollution and the COVID-19 pandemic mortality data by Poisson, geometric, and negative binomial regression models with fixed and random effects. *Open Chemistry*, 21(1), 20230364.

---

> ### Comment · Reviewer_ZRFo · 2023-11-22
> **Thanks for the reply**
>
> Thank you for your detailed reply and for conducting the additional experiments using a real-world example. These efforts have significantly enhanced my understanding and appreciation of your work. With these new insights, I am now more convinced of the value that your paper contributes to the field.

---

### Official Review · Reviewer_z2gb · 2023-11-01

**Soundness:** 3 good
**Presentation:** 2 fair
**Contribution:** 3 good
**Rating:** 6
**Confidence:** 3

**Summary:**

Here the authors presents a novel framework that merges hierarchical likelihood learning with Poisson-gamma Deep Neural Networks. This approach aims to better handle clustered count data by incorporating both fixed and random effects into the model. A proposed two-phase training algorithm is aimed at improving model performance. The proposed method is evaluated on both simulated data and real data where it is compared to existing methods, various types of DNNs and GLMs.

**Strengths:**

- The paper introduces a novel Poisson-gamma Deep Neural Network (DNN) and supports it with a robust mathematical foundation. This includes deriving h-likelihood functions and addressing identifiability issues, which demonstrates a high level of theoretical rigour.
- Incorporation of Random Effects: The model incorporates both fixed and random effects into a deep learning framework, aiming to capture both marginal and subject-specific variations in count data. This is particularly useful for applications where understanding both population and subject-specific trends is crucial such as in healthcare or social sciences.
- Two-Phase Training: The paper proposes a comprehensive two-phase training algorithm that focuses on both pretraining and fine-tuning of the model parameters. This adds to the model's could potentially improve its generalisation to unseen data.
- The motivation is well founded and I believe it would be of relevance to the ICLR community.

**Weaknesses:**

- The largest weakness is the unconvincing experimental result. They do not confirm fully confirm the advantages of the method (as claimed in the abstract). The distribution of the multiple runs on simulated data does not show obviously outperformance of this method compared to the others. It would have been interesting to see a statistical test of difference performed here.
- Related to the previous point, results on real datasets are presented without any replications or multiple runs. No errors are shown in table  2 making it very hard to judge the importance of the values. It should not be terrible expensive to perform multiple runs on these relatively small datasets. It would also have been very interesting to see this method applied to much larger datasets.
- The feature importance experiment (Figure 4) is valuable but I feel that it lacks context when not presented with more standard feature importance methods (e.g. permutation importance). I understand these are simulations with a pre-determined ground truth but contrasting this new method with a well-understood method would be a welcome addition to help better showcase the relative properties of this method.
- Although the two-phase training can bring benefits, as set out in the paper, it is also important to understand the cost that comes with it. A clear comment from the authors on the absolute and relative efficiency (or better, computational cost per training compared with other methods) would be a very useful addition from practical standpoint. Without one, the reader is left wondering if this truly is practical.

**Questions:**

- Spelling: e.g. in 4.2 "In contrast to HGLM and DNN" should be HGLMs and DNNs
- What is the early stopping method used for the experiments? Is it the same for pre-training and training?
- The method does not perform as well as PG-GLM on the fruits dataset but better on the others. Do the authors have any hypotheses as to why that is the case? What is different about these datasets?
- How do the authors envision the scalability of the proposed framework when dealing with extremely large datasets, particularly in medical and health applications?
- Could this method be extended to a larger framework to include other types of random effects distributions beyond the gamma distribution?

---

> ### Author Response · Authors · 2023-11-20
> **Response to Reviewer z2gb**
>
> We greatly appreciate your nice comments on our paper. We have prepared responses for your questions and revised our manuscript. We supplemented our experimental studies with increasing sample size in Section 5 and permutation importance in Section 4.4 and Appendix A.5. We provided p-values from Wilcoxon signed-rank test for comparing RMSPEs in experimental studies and real data analyses. We are pleased to strengthen our paper, and we hope it is satisfactory for you too.
>
> ### **Weakness 1 and 2**
>
> For a statistical test, we performed the Wilcoxon signed-rank test for the RMSPEs, which is widely-used for comparing the prediction errors [1-3].
>
> * In experimental studies, PG-NN outperforms the other models with p-value$<$0.001 in all scenarios except for G(0), with the absence of random effects. In this case, P-NN performs the best.
> * Following Tran et al. [4, Section 6.2.4], we did not consider the cross-validation for longitudinal data (Epilepsy, CD4, Bolus), because the last observations (test data) were predicted by training the past data for each subject. For clustered data (Owls, Fruits), we present the average of RMSPEs from 10-fold cross-validation with the p-values from Wilcoxon signed rank test. The following table (Table 3 in Appendix A.6) shows the mean RMSPEs and p-values for testing whether the RMSPE of PG-NN is less than that of other methods. (Hence the p-value for PG-NN itself is not included in this table.) In Owls data, though the p-values are marginal (0.05$\sim$0.1) in some cases, the proposed PG-NN significantly (p-value$<$0.05) outperforms the other methods. In Fruits data, the PG-NN has the smallest RMSPEs but the difference is not significant except for N-NN and NN-NN. Note here that the P-GLM was the best in Fruits data without cross-validation, but the cross-validation clarifies that the PG-NN has better overall RMSPE.
>
> |Model|Owls/RMSPE|Owls/p-value|Fruits/RMSPE|Fruits/p-value|
> |---|---|---|---|---|
> |P-GLM |2.493|0.032|6.203|0.065|
> |PN-GLM|6.695|0.001|5.953|0.138|
> |PG-GLM|6.689|0.001|5.954|0.138|
> |N-NN  |2.492|0.065|7.128|0.001|
> |NF-NN |2.602|0.080|6.159|0.042|
> |NN-NN |2.473|0.032|6.566|0.003|
> |P-NN  |2.463|0.042|6.234|0.053|
> |PF-NN |2.496|0.007|5.906|0.161|
> |PG-NN |2.427|     |5.901|     |
>
> We are willing to try with much larger real datasets, such as microbiome data and pandemic mortality data [5-9], but they were not open in public. Thus, in Section 4.4, we presented an experiment with $N=200,000$ and $p=100$. Average computing time for this experiment was about 30 minutes, which implies that the model can be applied to much larger datasets.
>
> ### **Weakness 3**
>
> We used the attention-based feature selection method to show that the proposed method can be adapted for various state-of-the-art neural network architectures. More standard feature importance methods can also be implemented to the proposed framework. Following your advice, we implemented the permutation importance method to our experiment in Section 4.4. Both the attention-based feature selection and permutation importance can identify the genuine features from the irrelevant features. The main difference between them is that the attention-based feature selection can achieve sparsity during training by using the sparsemax transformation, whereas the permutation importance is a post-hoc procedure which is calculated after the model has been fitted. Figures 7 and 8 in Appendix A.5 show the attention scores and permutation scores from one repetition, respectively. (Due to page limitations, Figure 4 in the initial submission was moved to Figure 6 in Appendix A.5, which shows the average of 100 repetitions to summarize all the results.)
>
> * [Figure 7 (Link)](https://i.imgur.com/Bya8aXy.png)
>
> * [Figure 8 (Link)](https://i.imgur.com/aAZASSR.png)
>
> ### **Weakness 4 and question 3**
>
> We presented an experiment with $q_{train}$ varying from 1 to 20 $(N=1,000 \sim 20,000)$. Figure 4 in Section 5 shows the average RMSPEs and computing times of standard PF-NN and the proposed PG-NN from 100 repetitions. The proposed PG-NN requires less computation time, while maintaining smaller RMSPE than standard PF-NN.
>
> * [Figure 4 (Link)](https://i.imgur.com/7YNF1xi.png)
>
> The experiment in Section 4.4 with N=200,000 and p=100 shows that the proposed method takes average 30min for each repetition. Furthermore, for handling extremely high-cardinality categorical features, it has been acknowledged that the random effect models have less degrees of freedom (less model complexity) than the fixed effect models [10, Chapter 6.5]. Thus, though we did not analyze such extremely large datasets, we expect that our framework is suitable to such large datasets.

---

> > ### Author Response · Authors · 2023-11-20
> > **Response to Reviewer z2gb (continued)**
> >
> > ### **Question 1**
> >
> > Thank you. We fixed it.
> >
> > ### **Question 2**
> >
> > For pretraining, we restricted the maximum number of epochs, without a specific early stopping method. For training, we stopped training if the h-likelihood is not improved until patience$=10\sim50$ epochs, depending on the batch size.
> >
> > ### **Question 4**
> > The proposed method can be extended to other distributions of random effects, if we find a specific scale $v_i^*$ for the h-likelihood. If not, we need a computational method, which could be computationally expensive. Thus, in Section 5, we showed the robustness of the gamma distribution by applying our method to normal random effects. Even if the distribution of random effects is misspecified, the proposed PG-NN performs good.
> >
> >
> > ### **References**
> >
> > [1] Zhang, X., Sugano, Y., Fritz, M., \& Bulling, A. (2015). Appearance-based gaze estimation in the wild. In *Proceedings of the IEEE conference on computer vision and pattern recognition*, 4511-4520.
> >
> > [2] Zhang, X., Sugano, Y., Fritz, M., \& Bulling, A. (2017). MIIGaze: Real-world dataset and deep appearance-based gaze estimation. *IEEE transactions on pattern analysis and machine intelligence*, 41(1), 162-175.
> >
> > [3] Xiong, Y., Kim, H. J., \& Singh, V. (2019). Mixed effects neural networks (menets) with applications to gaze estimation. In *Proceedings of the IEEE/CVF conference on computer vision and pattern recognition*, 7743-7752.
> >
> > [4] Tran, M. N., Nguyen, N., Nott, D., \& Kohn, R. (2020). Bayesian deep net GLM and GLMM. *Journal of Computational and Graphical Statistics*, 29(1), 97-113.
> >
> > [5] Saison, J., Maucort Boulch, D., Chidiac, C., Demaret, J., Malcus, C., Cotte, L., ... \& Ferry, T. (2015, April). Increased Regulatory T-Cell Percentage Contributes to Poor CD4+ Lymphocytes Recovery: A 2-Year Prospective Study After Introduction of Antiretroviral Therapy. In *Open Forum Infectious Diseases*, 2(2), ofv063.
> >
> > [6] Zhang, X., Mallick, H., Tang, Z., Zhang, L., Cui, X., Benson, A. K., \& Yi, N. (2017). Negative binomial mixed models for analyzing microbiome count data. *BMC bioinformatics*, 18, 1-10.
> >
> > [7] Dong, M., Li, L., Chen, M., Kusalik, A., \& Xu, W. (2020). Predictive analysis methods for human microbiome data with application to Parkinson’s disease. *PLoS One*, 15(8), e0237779.
> >
> > [8] Yirga, A. A., Melesse, S. F., Mwambi, H. G., \& Ayele, D. G. (2020). Negative binomial mixed models for analyzing longitudinal CD4 count data. *Scientific reports*, 10(1), 16742.
> >
> > [9] Iyit, N., \& Sevim, F. (2023). A novel statistical modeling of air pollution and the COVID-19 pandemic mortality data by Poisson, geometric, and negative binomial regression models with fixed and random effects. *Open Chemistry*, 21(1), 20230364.
> >
> > [10] Lee, Y., Nelder, J. A., \& Pawitan, Y. (2017). *Generalized linear models with random effects: unified analysis via H-likelihood*. CRC Press.

---

> > ### Comment · Reviewer_z2gb · 2023-11-23
> >
> > I thank the authors for responding to my points. In particular, the statistical tests provide much stronger evidence of the claims made. Including these results in the appendix is a positive addition to the paper.